# An insect-like mushroom body in a crustacean brain

**Gabriella Hannah Wolff[1], Hanne Halkinrud Thoen[2], Justin Marshall[2], Marcel E Sayre[3], Nicholas James Strausfeld[3]\***

[1]Department of Biology, University of Washington, Seattle, United States; [2]Sensory Neurobiology Group, University of Queensland, Brisbane, Australia; [3]Department of Neuroscience, School of Mind, Brain and Behavior, University of Arizona, Tucson, United States

**Abstract** Mushroom bodies are the iconic learning and memory centers of insects. No previously described crustacean possesses a mushroom body as defined by strict morphological criteria although crustacean centers called hemiellipsoid bodies, which serve functions in sensory integration, have been viewed as evolutionarily convergent with mushroom bodies. Here, using key identifiers to characterize neural arrangements, we demonstrate insect-like mushroom bodies in stomatopod crustaceans (mantis shrimps). More than any other crustacean taxon, mantis shrimps display sophisticated behaviors relating to predation, spatial memory, and visual recognition comparable to those of insects. However, neuroanatomy-based cladistics suggesting close phylogenetic proximity of insects and stomatopod crustaceans conflicts with genomic evidence showing hexapods closely related to simple crustaceans called remipedes. We discuss whether corresponding anatomical phenotypes described here reflect the cerebral morphology of a common ancestor of Pancrustacea or an extraordinary example of convergent evolution.
DOI: https://doi.org/10.7554/eLife.29889.001

**\*For correspondence:** flybrain@ neurobio.arizona.edu

**Competing interests:** The authors declare that no competing interests exist.

## Introduction

Mushroom bodies are paired centers first identified in the brains of Hymenoptera (*Dujardin, 1850*). They have been defined anatomically as a discrete neuropil composed of an abundance of parallel fibers originating from densely packed cell bodies, called globuli cells or Kenyon cells (*Schürmann F-W, 1973*). Together, the parallel fibers form a distinctive peduncle that divides into two or more columnar tributaries termed lobes. Input and output neurons intersect parallel fibers down the length of each lobe, partitioning the lobe into discrete territories (*Mobbs, 1982*; *Li and Strausfeld, 1999*). In most neopteran insects (those insects possessing wings that bend over the abdomen), parallel fibers provide systems of dendrites that form a distal cap- or cup-like domain called the calyx that receives afferents from olfactory neuropils and, in many species, from the optic lobes and gustatory centers as well. Calyces are absent in palaeopteran insects, such as odonates and Ephemeroptera (dragonflies, darters, mayflies), as well as in secondarily aquatic Neoptera such as diving beetles (*Strausfeld et al., 2009*). Additional characters conventionally attributed to mushroom bodies include the presence of inhibitory feedback pathways from proximal to more distal levels of the lobes and calyx (*Leitch and Laurent, 1996*). Mushroom bodies are further defined by enriched expression of proteins that are known to be essential for learning and memory functions in *Drosophila* (*Skoulakis et al., 1993*; *Skoulakis and Davis, 1996*; *Wang et al., 1998*). The exclusive location of the paired mushroom bodies is laterally in the protocerebrum (forebrain), as they are in another mandibulate group, Myriapoda, which includes centipedes and millipedes (*Wolff and Strausfeld, 2015*).

**eLife digest** With more than four million species, arthropods are the largest and most diverse group of animals on the planet and include, for example, crustaceans, insects and spiders. They are defined by their segmented bodies, hard outer skeletons and jointed limbs. All arthropods share a common ancestor that lived more than 550 million years ago. Exactly how this ancestral arthropod gave rise to the myriad species that exist today is unclear but we know that at some point the arthropod family tree split into branches, one of which went on to become the crustaceans. The crustacean branch then split again, giving rise to a line of descendants that would become the insects.

But although insects evolved from crustaceans, the brains of insects possess structures that those of crustaceans do not. Known as mushroom bodies, these structures help to form and store memories. Their absence in crustaceans has therefore been an enduring mystery. Wolff et al. now add a piece to the puzzle by showing that one group of modern-day crustaceans, the mantis shrimps, does in fact possess mushroom bodies.

By visualizing cells and pathways within the brains of mantis shrimps, and also a number of closely related species, Wolff et al. show that only these shrimps possess true mushroom bodies. However, some of the mantis shrimp's close relatives possess a few attributes of these structures. This suggests that mushroom bodies are evolutionarily ancient structures that arose in a common ancestor of insects and crustaceans, before being lost or radically modified in most of the crustaceans.

So why did this happen? Mantis shrimps are top predators with excellent vision that hunt over considerable distances, requiring them to evaluate and memorize complex features of their environment. These cognitive demands, which might not be shared by other crustaceans, may have led to the mantis shrimps retaining their mushroom bodies. Further research into the brains and behavior of the mantis shrimp may provide insights into how mushroom bodies construct memories of a complex sensory world.

DOI: https://doi.org/10.7554/eLife.29889.002

Mushroom body function has been attributed to learning and memory (*Heisenberg, 2003*). Currently, much of our knowledge about the roles of mushroom bodies comes from studies showing that convergence of populations of relay neurons from the antennal lobes onto sparse subsets of mushroom body Kenyon cells underlies multiple representations of odors (*Campbell et al., 2013*; *Perez-Orive et al., 2002*; *Lin et al., 2014*); that mushroom bodies mediate appetitive and aversive long-term memory (*Waddell, 2013*); and that they support allocentric and sequential place memory such as during trap-line foraging and obligate parasitoidism (*Mizunami et al., 1998*; *Montgomery et al., 2016*; *Farris and Schulmeister, 2011*).

Given that mushroom bodies have been identified in Myriapoda (centipedes and millipedes) and Chelicerata (arachnids, horseshoe crabs, and sea spiders) (*Wolff and Strausfeld, 2015*), it might be assumed that these centers would be ubiquitous across arthropods. However, that assumption is awkward because the paired higher centers in the lateral protocerebrum of crustaceans share few, if any, of the identifying characteristics of lobed mushroom bodies as described above. These paired centers are the domed hemiellipsoid bodies.

Particularly in malacostracan crustaceans, such as crayfish and shrimps, hemiellipsoid bodies may correspond functionally to mushroom bodies. In many species the hemiellipsoid bodies are subdivided into discrete lateral domains and levels (*Sullivan and Beltz, 2004*). Their neurons originate from a dense population of minute cell bodies situated at the dorsal and dorsolateral surfaces of the lateral protocerebra, like the perikarya of globuli cells associated with the insect mushroom body. However, only in one group of crustaceans, the Anomura (hermit crabs), has there been any description of enriched proteins involved in learning and memory that characterize insect mushroom body lobes. In the land hermit crab *Coenobita clypeatus*, these proteins are confined to discrete concentric layers arranged at intervals through the depth of the hemiellipsoid body. Planar arrangements of pre- and postsynaptic neurons in these strata are comparable to orthogonal networks observed in the mushroom body lobes of insects. Hence, the hemiellipsoid bodies of land hermit crabs are

interpreted as homologues of insect mushroom bodies (*Brown and Wolff, 2012*). But the question remains whether such circuits have evolved convergently, particularly because marine hermit crabs show no evidence of orthogonal circuitry, nor do species of the related orders, Astacidea (crayfish and lobsters) and Brachyura (true crabs). Nevertheless, crustaceans might be expected to have evolved insect-like mushroom bodies if, like insects, they engage in behaviors that demand high level sensory associations and memory (*Withers et al., 2008*), such as trap-line foraging (*Montgomery et al., 2016*), rational courtship (*Arbuthnott et al., 2017*), individual recognition, establishment and defense of territories, and obligate predation by stealth (*Libersat and Gal, 2013*; *Berens et al., 2017*; *Alcock and Bailey, 1997*).

Stomatopoda (mantis shrimps), which is the sister group of all eumalacostracan crustaceans (*Wolfe et al., 2016*), engage in such behaviors. Stomatopods possess the most elaborate visual systems of any arthropod, where each eye moves independently of the other, and is endowed with binocular vision and numerous color- and polarization-sensitive channels, including channels for detecting circular polarization (*Marshall et al., 2007*). Mantis shrimps frequent established hunting territories distant from their lair (*Cronin et al., 2006*). Like insects, mantis shrimps form associative memories of kin and place through sensory integration employing the modalities of vision and olfaction (*Caldwell and Lamp, 1981*) and can be trained in discrimination-learning paradigms using color and polarized light stimuli (*Marshall et al., 1999*). Individuals remember conspecifics with which they have had aversive or non-aversive encounters (*Caldwell, 1992*). These behaviors require a brain equipped for memory of places, territories and individual recognition (*Marshall et al., 1996*). Here we report the first comprehensive description of centers in Stomatopoda that correspond in detail to the mushroom bodies of insects.

Their similarity of brain structures has suggested phylogenetic proximity of insects and malacostracan crustaceans (*Strausfeld and Andrew, 2011*). However, genomic evidence argues strongly that insects are instead most closely related to blind, morphologically simple cave-dwelling crustaceans, called remipedes (*Regier et al., 2005*; *Oakley et al., 2013*; *von Reumont et al., 2012*; *Schwentner et al., 2017*), which like all crustaceans other than stomatopods lack mushroom bodies. The identification of mushroom bodies in mantis shrimps sheds new light on insect-crustacean relationships. We show that, within more recently radiating eumalacostracan lineages, the neuronal composition and organization of mushroom body morphology appear to have undergone radical modification resulting in centers unrecognizable as mushroom bodies but retaining the function of a sensory integrator. The earlier identification of a hexapod-malacostracan-like brain in a fossil stem arthropod (*Ma et al., 2015*) together with the present results argues for an elaborate ancestral cerebral organization already present in the lower Cambrian.

## Results

### Identification of mushroom bodies in crustaceans

We have surveyed a broad range of Crustacea, particularly Malacostraca, for evidence of centers that would correspond to mushroom bodies identified in insects. Surprisingly, as outlined above, there have been very few criteria employed for identifying insect mushroom bodies. Those criteria are the presence of lobes, with or without an outer 'calyx' or 'cap', arising from many hundreds of minute cell bodies clustered laterally in the forebrain (*Flögel, 1878*). Original descriptions of mushroom bodies emphasized a distal dendritic elaboration, called a calyx, although subsequent studies have shown that many insect groups lack this, as do Myriapoda and Chelicerata (*Wolff and Strausfeld, 2015*).

Surprisingly, just these few characters have sufficed historically for claiming homology of lobed centers across Hexapoda. However they are hardly adequate to claim homology of neuropils in divergent arthropod lineages such as the two sub-phyla, Crustacea and Hexapoda, that comprise the clade Pancrustacea. Here we expand neuroanatomical criteria to unique characters that we view as necessary and sufficient to claim a center's identity as a mushroom body and hence support possible phenotypic homology across species. *Table 1* lists these characters and their occurrence in representative insects (*Drosophila* and the cockroach *Periplaneta americana*) and, within eumalacostracan crustaceans, Stomatopoda and five decapod species. Here Stomatopoda are

**Table 1.** Matrix of characters 1–13 defining insect-type mushroom bodies.

'1' denotes character presence, '0' denotes character absence, and "– "denotes lack of data. Character list. 1. Density of dorsal cluster of globuli cells exceeds $3.0 \times 10^6/mm^3$. 2. Distal calycal/domed (synonymous) neuropil. 3. Distal microglomeruli. 4. Spiny globuli cell dendrites. 5. Clawed globuli cell dendrites. 6. Globuli cells lacking dendrites. 7. Parallel fibers. 8. Orthogonal networks. 9. Efferent dendritic trees intersecting columns/lobes. 10. Aminergic afferents. 11. Afferent/efferent processes partitioned into domains. 12. GAD/GABAergic recurrent pathways. 13. Elevated expression of proteins required for learning and memory. Examples of characters. For character 1–11, see *Figure 1—figure supplement 1*; for example of character 12 see *Figure 3B,C*; for examples of character 13 see *Figure 4* and *Figure 5—figure supplements 1* and *2*. Species list. Neopteran insects: *Drosophila melanogaster*, *Periplaneta americana*, Odonata (dragonflies, darters): *Perithemis tenera*, *Libellula depressa*, *Aeschna* sp.). Stomatopod crustaceans: *Neogonodactylus oerstedii*, *Gonodactylus smithii*. Decapod crustaceans: *Alpheus bellulus*, *Stenopus hispidus*, *Procambarus clarkii*, *Coenobita clypeatus*, *Hemigrapsus* (*H. nudus* and *H. oregonensis*).

|  | 1 | 2 | 3 | 4 | 5 | 6 | 7 | 8 | 9 | 10 | 11 | 12 | 13 |
|---|---|---|---|---|---|---|---|---|---|---|---|---|---|
| *D. melanogaster* | 1 | 1 | 1 | 1 | 1 | 0 | 1 | 1 | 1 | 1 | 1 | 1 | 1 |
| *P. americana* | 1 | 1 | 1 | 1 | 1 | 0 | 1 | 1 | 1 | 1 | 1 | 1 | 1 |
| Odonata | 1 | 0 | 0 | 0 | 1 | 1 | 1 | 1 | 1 | 1 | 1 | – | 1 |
| Stomatopoda | 1 | 1 | 1 | 1 | 1 | 1 | 1 | 1 | 1 | 1 | 1 | 1 | 1 |
| *A. bellulus* | 1 | 1 | – | 1 | 1 | 0 | 1 | 1 | – | – | – | – | 1 |
| *S. hispidus* | 1 | 1 | – | – | – | – | 1 | 1 | 0 | – | – | – | 1 |
| *C. clypeatus* | 1 | 1 | – | 0 | 1 | 0 | 0 | 1 | 0 | – | 1 | – | 1 |
| *P. clarkii* | 1 | 1 | 1 | 0 | 1 | 0 | 0 | 0 | 0 | – | 0 | 0 | 0 |
| *Hemigrapsus* | 0 | 1 | – | 0 | 1 | 0 | 0 | 0 | 0 | 0 | 0 | 0 | 0 |

DOI: https://doi.org/10.7554/eLife.29889.007

represented by two species, *Gonodactylus smithii* (*Figure 1A,B*) and *Neogonodactylus oerstedii*. Examples of these characters are shown in *Figure 1—figure supplement 1*.

## Identification of mushroom bodies in Stomatopoda

In stomatopods, as in most stalk-eyed crustaceans, the brain's lateral protocerebra are contained within the distal eyestalks (*Figure 1C*). These are exceptionally capacious in mantis shrimps due to the voluminous nested optic lobe neuropils, which serve the largest and most complex compound eyes yet described (*Figure 1A*). The eyestalk also accommodates elaborate neuropils that in the literature have been collectively referred to as the 'medulla terminalis.' These neuropils comprise the lateral protocerebrum, just as they do in insects (*Figure 1C*, *Figure 1—figure supplement 2*). In the mantis shrimp, the largest of these neuropils originates from a dorsal cell body cluster of about 170,000 (in *N. oerstedii*) minute and tightly clustered neuronal perikarya, which we here refer to as globuli cells. This population provides the neurons that form four columnar neuropils, each comprising thousands of parallel fibers. Only one of the four columns is surmounted by a large calyx-like neuropil (*Figure 1D*) composed of layers of dendritic arborizations. The four columns (*Figure 1E*), which extend latero-caudally, together demarcate the proximal margin of the eyestalk neuropil, near its junction with the eyestalk nerve (*Figure 1C*).

Globuli cells provide one very large dome-like neuropil (here termed the calyx, indicated Ca in *Figure 1D,E*) from which one of the smaller columns originates (*Figure 1—figure supplement 3A*). Globuli cells also provide parallel fibers constituting three additional columns (*Figure 1E*; *Figure 2B, C*), one of which is equipped with a small calyx distally (*Figure 1—figure supplement 3B*). The two other columns lack distal dendrites entirely. Neurites from globuli cells converge at the origin of each column, as they do at the pedunculus (or initial stalk) of an insect mushroom body (*Figure 1—figure supplement 3A,B*). Each cell body neurite then increases in girth and extends for hundreds of microns as a thicker process decorated with varicosities and spines representing, respectively, pre- and postsynaptic sites (*Strausfeld and Meinertzhagen, 1998*). Like Kenyon cells in insect mushroom bodies, globuli cells in stomatopods lack a proper axon: each globuli cell's parallel fiber serves as a local interneuron, corresponding in shape and decoration to the parallel fibers of the insect mushroom body lobes, exemplified by *D. melanogaster* (*Figure 1F*). And by comparison to the

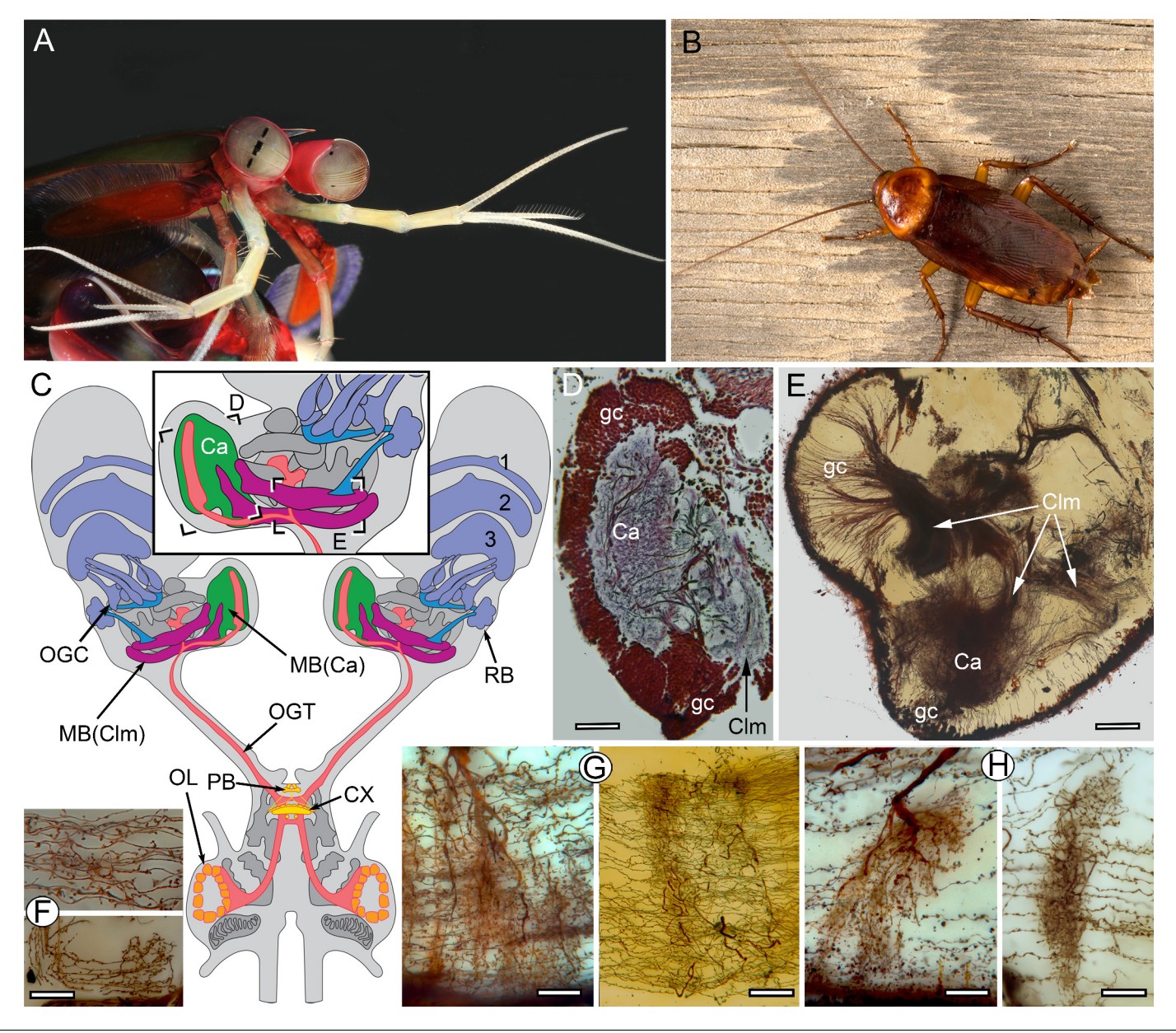

**Figure 1.** The stomatopod brain and mushroom body organization common to stomatopods and insects. (**A**) Eyestalks and antennules of *Gonodactylus smithii*. With the largest compound eyes of any panarthropod, and long antennules, mantis shrimps are equipped for high-level visual and olfactory perception. Information from both sensory systems reaches the mushroom body calyces and its column-like extensions. (**B**) *Periplaneta americana*, the American cockroach provides the hexapod species for comparisons with the stomatopod. (**C**) The central forebrain of the stomatopod *Gonodactylus smithii* is defined by the central complex (CX) and associated protocerebral bridge (PB). This contrasts with the forebrain's enormous lateral protocerebra in which optic neuropils and mushroom bodies (MB) dominate. Bifurcating axon bundles of the main tract from the olfactory lobes (OL) to the protocerebrum, called the olfactory-globular tract [OGT shown orange] supply the mushroom body calyces [MB(Ca), green]. Proximal to the optic lobes, units of the optic glomerular complex (OGC) and the reniform body (RB) send relays to the mushroom body columns [MB(Clm), magenta]. Boxed areas in the inset, show the locations of panels *D-F*. (**D**) Silver-stained calyx (Ca) associated with just one of four columns (Clm) is shown with the dense population of globuli cells (gc) that supply all four columns. (**E**) Golgi impregnation resolving the calyx (Ca) and three of the four mushroom body columns (Clm) originating from the globuli cell layer (gc). (**F**) Parallel fibers decorated by beaded or spiny arborizations indicative of pre- and postsynaptic sites (*Strausfeld and Meinertzhagen, 1998*) are a defining feature of insect mushroom bodies. Here parallel fibers in *G. smithii* (upper panel) are compared to those in the α'-lobe of *Drosophila melanogaster* (lower panel). (**G**) Regions along mushroom body columns in *G. smithii* (left) and the mushroom body lobes of the cockroach *P. americana* (right) show corresponding orthogonal networks of parallel fibers intercepted by the arborizations of afferent (input) neurons. (**H**) Parallel fibers intercepted by dendritic trees of output (efferent) neurons in a mushroom body column of *G. smithii* (left) and a lobe of *P. americana* (right). Scale bars: b, c, 100 μm; d-f, 20 μm.

*Figure 1 continued on next page*

*Figure 1 continued*

DOI: https://doi.org/10.7554/eLife.29889.003

The following figure supplements are available for figure 1:

**Figure supplement 1.** Examples of characters 1–13 shared by insect (*Periplaneta americana*, (P) and stomatopod (*Neogonodactylis oerstedii*, (N).

DOI: https://doi.org/10.7554/eLife.29889.004

**Figure supplement 2.** Corresponding topologies of insect and stomatopod cerebral regions.

DOI: https://doi.org/10.7554/eLife.29889.005

**Figure supplement 3.** Mushroom body calyces in *Neogonodactylus oerstedii*.

DOI: https://doi.org/10.7554/eLife.29889.006

*Drosophila* mushroom body lobe, each parallel fiber is assumed to contribute to local computational circuits within its column (*Cervantes-Sandoval et al., 2017*). Comparing the columns of the stomatopod mushroom body and the mushroom body lobes of the cockroach *Periplaneta americana* (*Figure 1G,H*) further demonstrates that parallel fibers in both species are regularly intersected by the branched processes of efferent and afferent neurons. As in the insect mushroom body, in the stomatopod mushroom body orthogonally organized neurons and local circuits provided by parallel fibers (*Figure 1H*) offer many thousands of possible sensory associations (*Heisenberg, 2003*; *Cassenaer and Laurent, 2007*; *Huerta et al., 2004*; *Aso et al., 2014a*).

In insects, the packing density of globuli cells associated with a mushroom body exceeds the packing density of neuronal cell bodies elsewhere in the brain. Our measurements show that the packing density of globuli cells supplying the stomatopod mushroom body is comparable to that of globuli cells supplying the mushroom bodies of insects, here exemplified by *P. americana* (*Figure 2A*). An important identifier of mushroom bodies in insects is the morphology of its Kenyon cells and their specializations; in particular, clawed specializations that partially wrap around large terminal boutons of sensory afferents in the calyces of flies and other insects (*Yasuyama et al., 2002*). These are also identified in stomatopods, but only on dendrites contributing to the smaller of the two calyces (*Figure 2D*). In the larger calyx, spined dendritic processes correspond to the spined dendrites of the mushroom body calyx of *P. americana* (*Figure 2A*). However, the stomatopod calyx is deeper than is an insect calyx and is composed of four distinct levels, the second of which is characterized by hundreds of thousands of synaptic clusters comparable to microglomeruli identified in the calyces of insects (*Figure 3A*).

In insects, the processes of uniquely identifiable efferent and afferent neurons are arranged at specific loci along the length of a mushroom body's lobe, giving the lobe the appearance of being segmented into discrete synaptic domains (*Li and Strausfeld, 1999*; *Ito et al., 1998*). In *Drosophila*, such neurons at specific loci along the lobes have been shown to encode the adaptive values of sensory information associated with learned responses (*Aso et al., 2014b*). The same type of organization typifies the stomatopod mushroom body columns (*Figure 2E,F*). And, as in the insect mushroom body, this arrangement is further demonstrated by the organization of aminergic neurons, the processes of which invade the columns at specific sites. In *N. oerstedii* GAD-immunoreactive neurons extending across the two largest columns are representative of centrifugal pathways back to the calyx (*Figure 3B,C*). Other aminergic afferents are resolved using antisera against serotonin (5HT) and tyrosine hydroxylase (TH). These also demonstrate discrete territories along the length of the mushroom body columns (*Figure 3F,G*). These arrangements of TH-immunoreactive neurons, first identified in the mushroom body lobes of the fly *Calliphora erythrocephala* (*Nässel and Elekes, 1992*), correspond to discrete dopaminergic territories in the mushroom body of *D. melanogaster*. Thermogenetics, optogenetics, and mutant analysis have demonstrated the pivotal role of these neurons in appetitive and aversive conditioning (*Waddell, 2013*; *Waddell, 2016*).

Three-dimensional reconstructions of the stomatopod mushroom bodies confirm that only one of the four columns (Clm 3, *Figure 2B*) originates from the extensive volume of dendritic processes that constitute the elaborate calyx (*Figure 2B*). Golgi impregnations resolve a smaller column (Clm 4) capped by a much smaller cluster of outer dendrites constituting a second calyx. The remaining two columns (Clm 1, 2) are the largest and are composed of 'naked' globuli cells entirely lacking distal dendrites (*Figure 1—figure supplement 3B*). That all four columns are supplied by globuli cells is another correspondence with the organization of the insect mushroom body, which develops from four cell lineages, the clonal progeny of which do not remain separated but merge to form a single

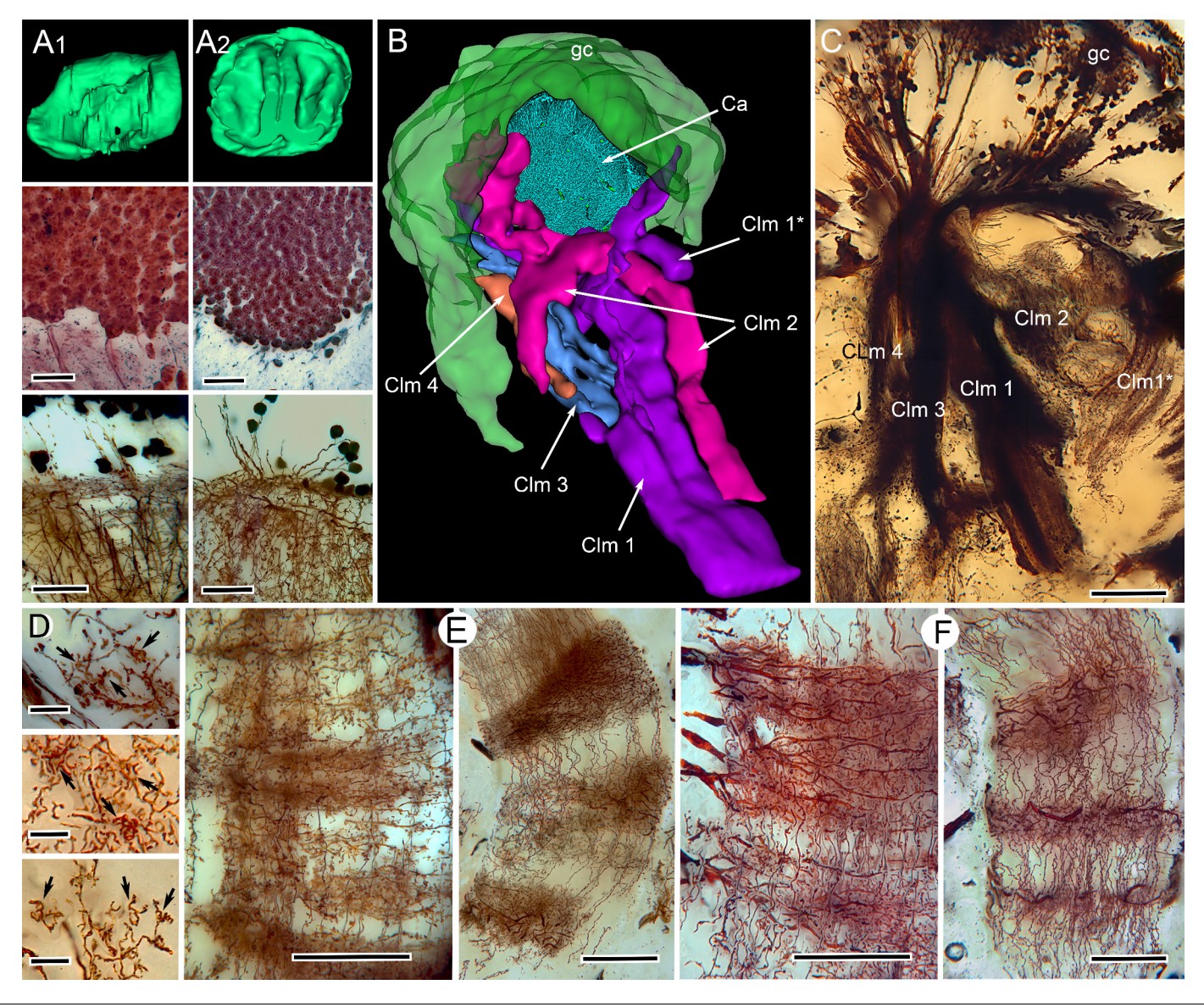

**Figure 2.** Correspondences of stomatopod and insect mushroom bodies. (*A1, A2 upper panels*) Globuli cell domains reconstructed from serial sections from *Neogonodactylus oerstedii* (**A1**) and *Periplaneta americana* (**A2**). (*A1, A2 middle panels*): Globuli cell packing (reduced-silver-stained sections). (*A1, A2 lower panels*) Golgi impregnations showing corresponding neuronal organization in calyces. (**B**) Three-dimensional reconstruction from serial sections of *N. oerstedii*, stained by reduced silver, demonstrates a mushroom body consisting of a globuli cell layer (gc, green), a calyx (Ca, cyan) and four distinct columns (Clm 1–4, purple, magenta, blue, orange), with Clm 1 accompanied by distal branch (Clm 1*). (**C**) Golgi impregnation of the columns, each defined by its characteristic arrangement of parallel fibers. (**D**) Clawed specializations (arrowed) are mushroom body identifiers found on dendrites arising distally from parallel fibers. These are shown here in the stomatopod (upper panel), cockroach (middle panel) and honeybee (lower panel). (**E, F**) The dense spiny processes of efferent (output) neurons form discrete domains along the length of the columns. Paired corresponding arrangements are here demonstrated from the stomatopod *N. oerstedii* (left in each pair) and *P. americana*. Scale bars: A, panels, second row, 25 μm; lower panels, 20 μm; C, 100 μm; D, scales in all panels, 10 μm; E-F, all scales = 50 μm.

DOI: https://doi.org/10.7554/eLife.29889.008

The following videos are available for figure 2:

**Figure 2—video 1.** Three dimensional reconstruction of the stomatopod mushroom body. Serial section reconstruction demonstrates the dispositions and relative sizes of mushroom body columns originating from the dorsal globuli cell cluster.

DOI: https://doi.org/10.7554/eLife.29889.009

**Figure 2—video 2.** Three dimensional reconstruction of the stomatopod mushroom body. Serial section reconstruction demonstrates the dispositions and relative sizes of mushroom body columns originating from the dorsal globuli cell cluster.

DOI: https://doi.org/10.7554/eLife.29889.010

neuropil (*Ito et al., 1997*; *Farris et al., 2004*). In the mantis shrimp, four distinct columns originate from the same globuli cell population but remain separate from each other (*Figure 2B*, *Figure 1—figure supplement 1*). The absence of a cap or calyx crowning two of the stomatopod mushroom body columns is comparable to the organization of mushroom bodies in palaeopteran insects (mayflies, damselflies), which entirely lack calyces (*Strausfeld et al., 2009*). This 'calyxless' condition also typifies mushroom bodies in Myriapoda, another mandibulate taxon (*Wolff and Strausfeld, 2015*). Ancestral morphology is also suggested by mushroom body development in insects, where outgrowth of the lobes precedes the development of the calyx (*Ito et al., 1997*; *Farris et al., 2004*).

To summarize, the combination of neuroanatomical characters that distinguish the insect mushroom body also distinguish the stomatopod mushroom body (*Table 1*). Furthermore, the stomatopod mushroom body columns and the insect mushroom body lobes are recognized at the molecular level by antibodies raised against three proteins required for learning and memory in *Drosophila*: (1) protein kinase A catalytic subunit alpha (DC0) (*Skoulakis et al., 1993*), (2) Leonardo (Leo), the ortholog of mammalian 14-3-3ζ (*Skoulakis and Davis, 1996*), and (3) phosphorylated calcium/calmodulin-dependent protein kinase II (pCaMKII) (*Wang et al., 1998*). In the cockroach lateral protocerebrum, all three antibodies selectively resolve the mushroom body lobes (*Figure 4F*). Similarly, in the lateral protocerebrum of the stomatopod brain these three antibodies selectively resolve all four columns identified by reduced silver staining and Golgi impregnation (*Figure 4B–E*).

## Divergent evolution of the mushroom bodies

Mapping brain morphologies onto a representative genomic phylogeny of Pancrustacea (*Oakley et al., 2013*) demonstrates the absence of mushroom bodies in all taxa intervening between the clade Hexapoda+Remipedia and Eumalacostraca (*Figure 5A*). To examine whether mushroom bodies may have evolved divergent arrangements within Eumalacostraca (*Figure 5A*), we analyzed five species of more recently evolved groups of decapods (*Shen et al., 2013*): Caridea (including snapping or pistol shrimps), Stenopodidea (cleaner shrimps), Astacidea (including lobsters and crayfish), Anomura (land hermit crabs), and Brachyura (true crabs; *Figure 5B*). Alpheid caridids ('pistol shrimps') are the only group known to have evolved eusociality. Stenopid ('cleaner shrimps') hemiellipsoid bodies have been previously described, published images suggesting a short extension from a large hemiellipsoid body (*Sullivan and Beltz, 2004*). Comparing the DC0 immunocytology of these taxa with that of the *Drosophila* mushroom bodies (*Figure 5—figure supplement 1A*) identifies DC0-positive structures in two of the investigated species. In the banded coral shrimp *Stenopus hispidus* (Stenopodidea), anti-DC0 resolves a pedestal-like extension from its domed hemiellipsoid body (*Figure 5—figure supplement 1B*), which is divided into three distinct territories. Anti-DC0 also reveals prominent mushroom body-like centers in *Alpheus bellulus* (Caridea). Golgi impregnations of *A. bellulus* show these prominently in the lateral protocerebrum where they are of comparable size to those of social or parasitoid hymenopteran insects (*Figure 5—figure supplement 1C,D*).

The land hermit crab *C. clypeatus* (Anomura) is distinct: its hemiellipsoid body consists of an enormous domed neuropil that is characterized by concentrically arranged DC0- and pCaMKII-immunoreactive layers (*Figure 5—figure supplement 2A*). Each is a flat sheet comprising a planar arrangement of globuli cell processes that are orthogonally intersected by the arborizations of input and output neurons. Although these arrangements are peculiar to land hermit crabs, they demonstrate that the network organization within each layer corresponds to the three-dimensional networks that are defined by the sequential domains of input and output arborizations that extend across parallel fibers in the insect mushroom body (*Brown and Wolff, 2012*; *Wolff et al., 2012*). Only in one group outside Malacostraca, is there a neuropil at all reminiscent of a mushroom body. This is in Cephalocarida, where an allantoid center, occupying an otherwise drastically reduced protocerebrum, receives inputs from the olfactory lobes on the same side of the brain, as occurs in insects (*Stegner and Richter, 2011*).

In crayfish (Astacidea), the dorsal globuli cell cluster provides a prominent domed center in the lateral protocerebrum. However this hemiellipsoid body does not show discernable levels of the proteins involved in *Drosophila* learning and memory (*Figure 5—figure supplement 2C*). The hemiellipsoid bodies of the examined varunid crabs (Brachyura) are inconspicuous and they too lack elevated levels of those proteins (*Figure 5—figure supplement 2B*). Together, with other neuroanatomical criteria (*Table 1*), these observations suggest that more recently evolved Eumalacostraca show a reduction or complete loss of mushroom body-like lobes (columns), accompanied by accentuation

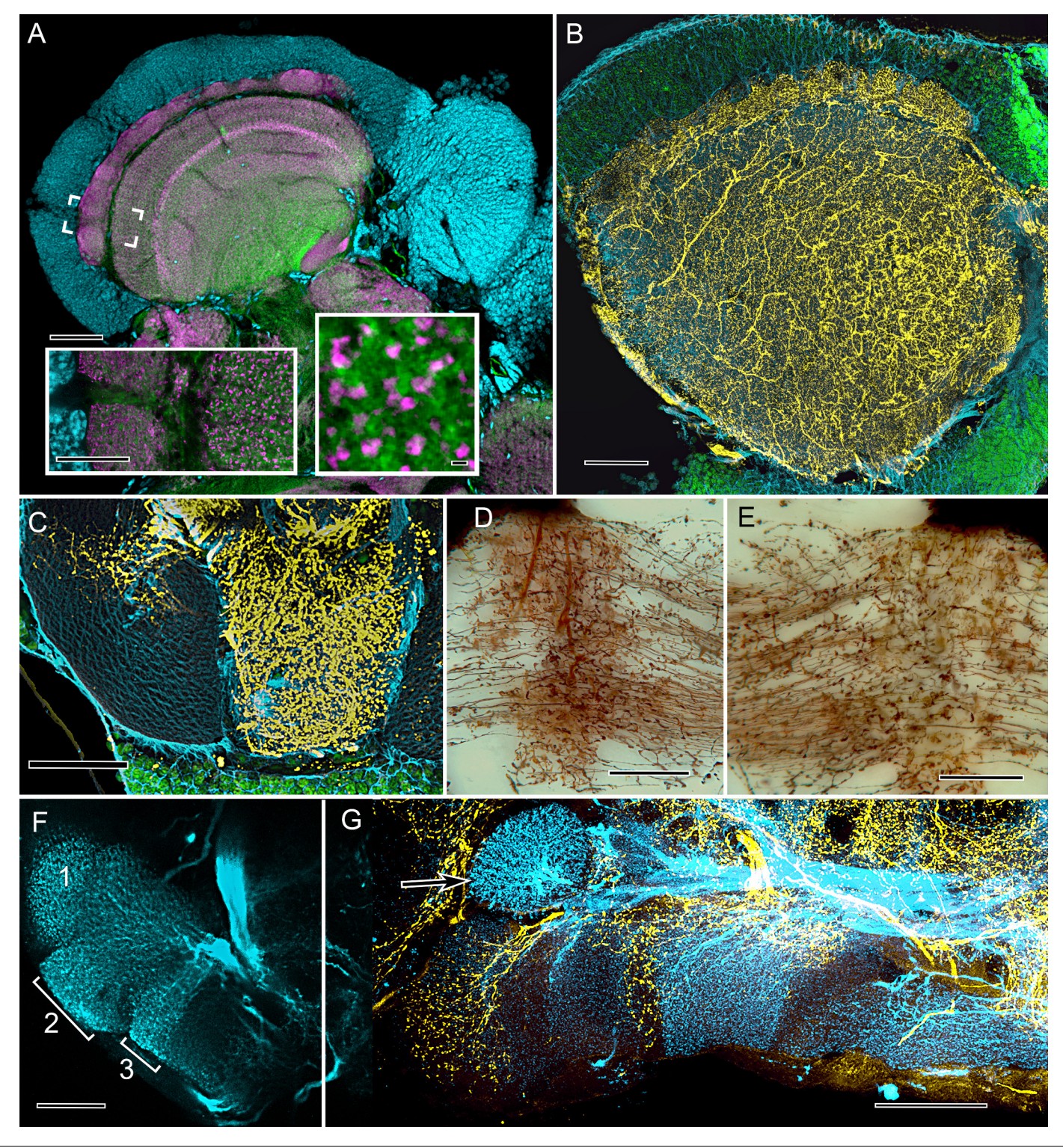

**Figure 3.** Cardinal features of the stomatopod calyx and neuromodulatory neurons in MB columns. (**A**) Double Immunolabeling with actin-phalloidin (green), anti-synapsin (magenta), and the fluorescent nuclear stain Syto-13 (blue) resolves globuli cells and the concentric synaptic layers in the calyx of *Neogonodactylus oerstedii*. *Brackets* indicate the location of the *left inset*, an enlargement of the outer two layers, the deeper of which is characterized by synaptic microglomeruli (enlarged in *right inset*), which correspond to those in insect calyces. (**B**) Top-down view of a 60 µm section cut tangential to the stomatopod calyx labeled with anti-GAD (yellow), anti-α-tubulin (blue) and Syto-13 (green). (**C**) Part of mushroom body column 2 showing a dense domain of GAD-immunoreactive processes (yellow) with a smaller arborization upper left. (**D**) Afferent terminal in column 2 with profusely decorated
*Figure 3 continued on next page*

*Figure 3 continued*

varicosities possibly corresponding to those of the anti-GAD labeled arborization in (**C**). (**E**) An intrinsic orthogonal network provided by branches from parallel fibers. (**F**) Anti-TH whole-mount labeling shows "segmentation" into three afferent domains . (**G**) Double-labeling with anti-5HT (yellow) and anti-TH (blue) shows segmentation of column 1 by domains occupied by serotonin and tyrosine hydroxylase. A single-TH afferent, shown as a transverse profile, reveals column 3 in cross section (arrow, left). Scale bars A and loer left inset 100 µm, inset lower right 1 µm. C, 100 µm; C-F, 50 µm; G, 100.

DOI: https://doi.org/10.7554/eLife.29889.011

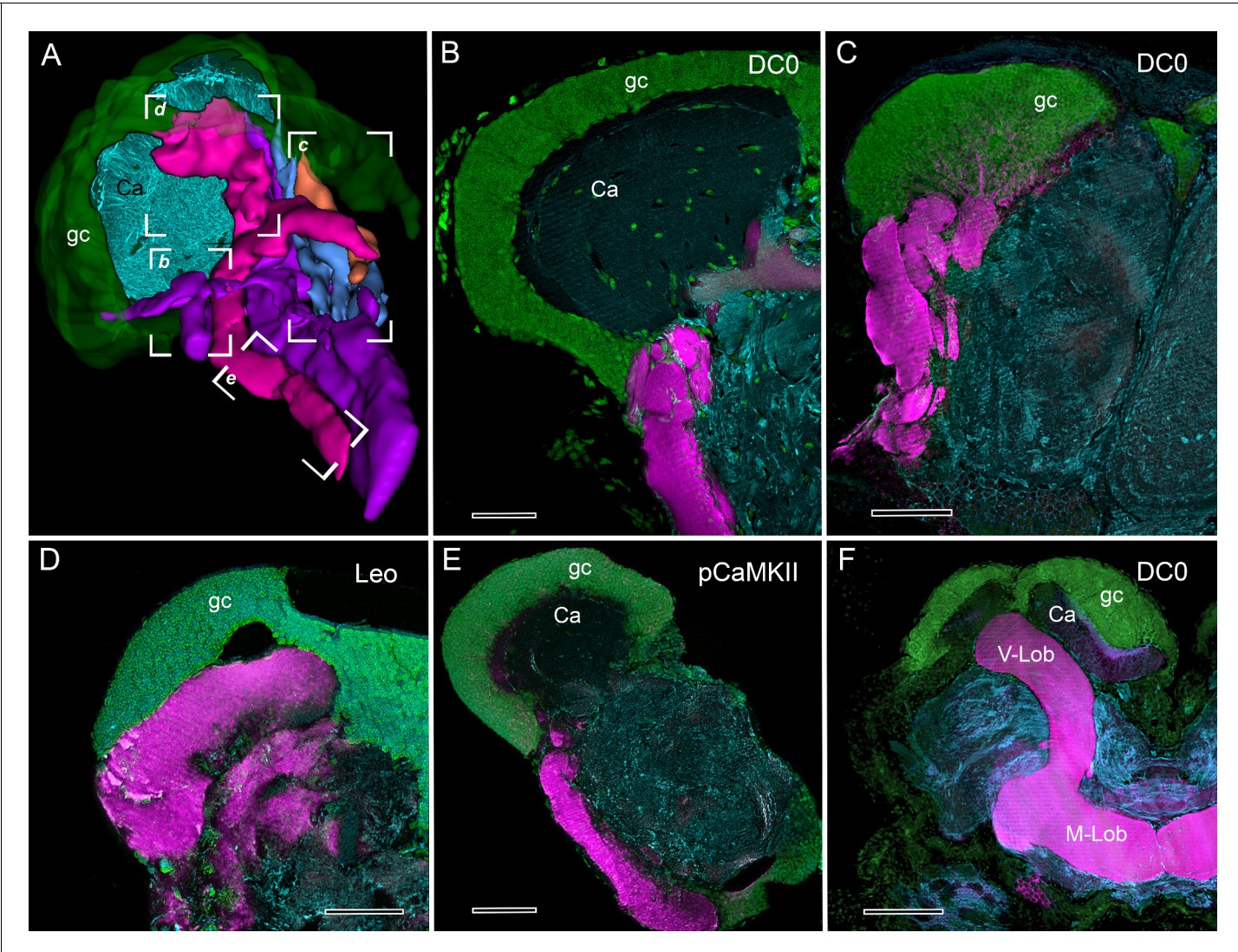

**Figure 4.** Selective affinity of stomatopod and insect mushroom bodies to antisera against proteins involved in *Drosophila* learning and memory. (**A**) Reconstruction from reduced-silver serial sections of the mushroom body of *Neogonodactylus oerstedii* (globuli cell layer, green; calyx, cyan; columns are shown purple, magenta, blue, orange). Brackets indicate panels *B-E* showing corresponding regions of the columns. (**B–E**) Mushroom body columns in *N. oerstedii* labeled by anti-DC0 in magenta (**B, C**). (**D**) Labeling for Leo (magenta). (**E**) Labeling for pCaMKII (magenta). (**F**) Mushroom body lobes of *Periplaneta americana* labeled with antibodies against DC0 in magenta and showing globuli cells (gc), the underlying calyx (Ca) and vertical and medial lobes (V-Lob, M-Lob). In panels *B-F*, globuli cells are labeled green with the nucleic acid stain, Syto-13; antibodies against α-tubulin (cyan) reveal background structure. Scale bars: 100 µm.

DOI: https://doi.org/10.7554/eLife.29889.012

and diversification of domed neuropils that have been historically referred to as hemiellipsoid bodies (*Figure 5B*).

## The reniform body

Other elongated centers exist within eyestalk neuropils. One that is conspicuous in Stomatopoda, and which has an equivalent in the Varunidae (shore crabs belonging to Brachycera), offers an interesting counterpoint to the identification of the stomatopod mushroom body. This is the reniform body, a prominent compound neuropil located near the inner margin of the optic lobes (RB in *Figure 1C*), which in *N. oerstedii* arises from a dense cluster of about a thousand cell bodies. These

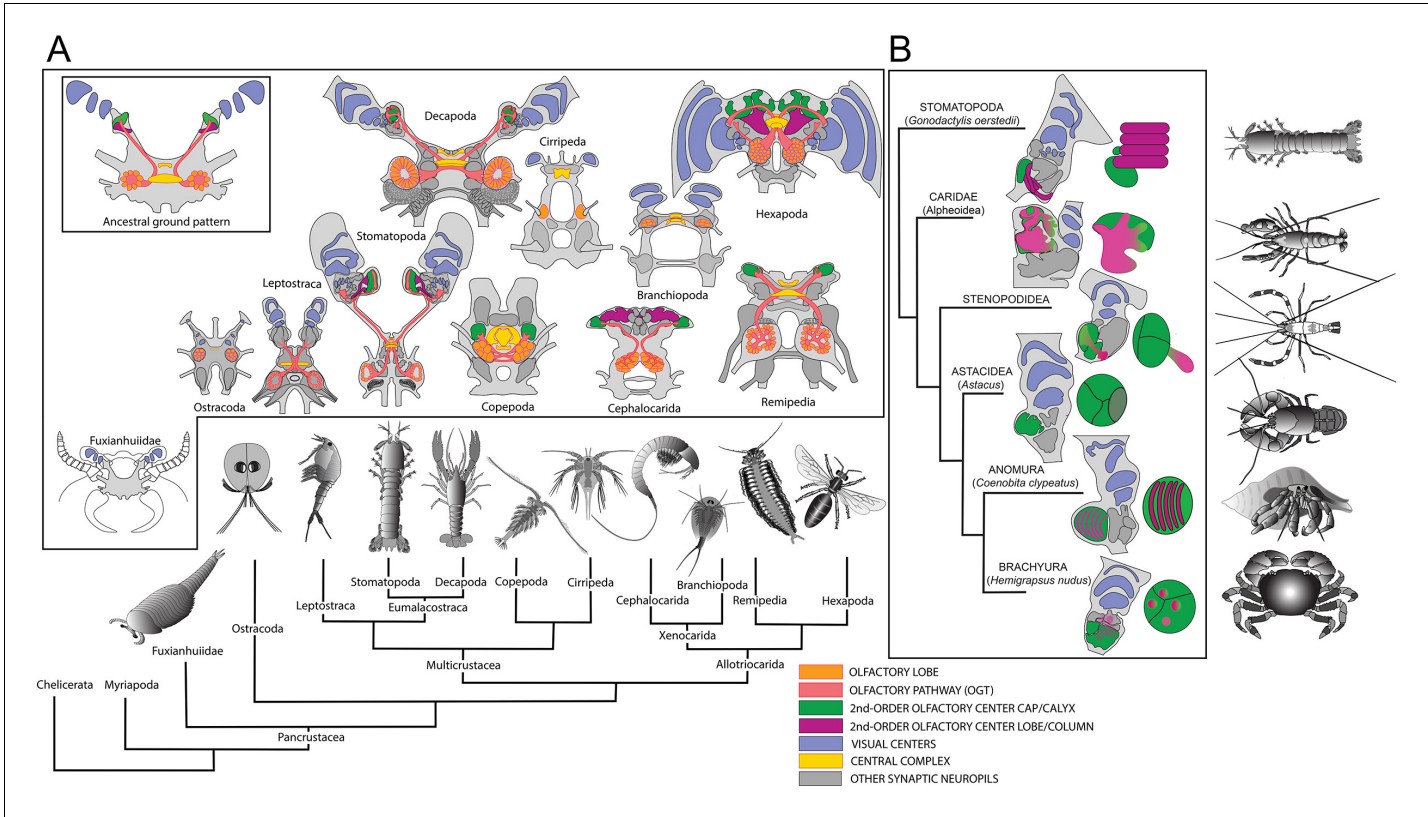

**Figure 5.** Molecular phylogeny (from *Oakley et al., 2013*) showing the pattern of derived cerebral arrangements across Pancrustacea (Crustacea +Hexapoda) with corresponding organization in Hexapoda and Stomatopoda. (**A**) Molecular phylogenetics resolves Hexapoda more closely related to Remipedia than either is to Malacostraca (see *Oakley et al., 2013*). In insects and stomatopods mushroom bodies share all the key characters identified in this study, and the brains of these two groups possess other common organization: three nested optic neuropils, fusion of three brain neuromeres – all identified as a ground pattern in the lower Cambrian euarthropod *Fuxianhuia protensa* (*Ma et al., 2012*) – as well as corresponding central complexes. These features define the ancestral ground pattern of the pancrustacean brain. In contrast, all other brains have evolved divergent modifications including reduction and loss of some or all of these ancestral components. (**B**) Phyletic positions of five eumalacostracan sister groups to Stomatopoda showing modifications of the ancestral columnar mushroom body present in Stomatopoda. In representative species belonging to these lineages, column-like extensions from globuli cells have become enormously enlarged (Alpheoidea), greatly reduced but still indicated by DC0 immunoreactivity (Stenopodidea), subsumed as discrete layers within the hemiellipsoid body (Anomura), or lost entirely (Brachyura and Astacidea). Confocal images of each example, comparing these with the lobed organization of the *Drosophila* mushroom body, are shown in *Figure 5—figure supplements 1* and *2*. In Stenopodidea and crownward, the calyx observed in Stomatopoda assumes the morphology of a hemiellipsoid body.
DOI: https://doi.org/10.7554/eLife.29889.013

The following figure supplements are available for figure 5:

**Figure supplement 1.** Mushroom body lobes are recognized by their DC0 immunoreactivity across pancrustaceans.
DOI: https://doi.org/10.7554/eLife.29889.014

**Figure supplement 2.** Divergence of decapod hemiellipsoid bodies.
DOI: https://doi.org/10.7554/eLife.29889.015

**Figure supplement 3.** Stability of brain organization over time.
DOI: https://doi.org/10.7554/eLife.29889.016

provide a pedestal-like arrangement of smooth parallel neurites supplying four domains: an initial zone and a lateral, distal, and proximal zone. It is the latter three that define the center's characteristic kidney shape.

All four zones contain the dendritic processes of collaterals branching from the pedestal as well as numerous branched afferent processes that supply them (*Figure 6A,B*). Antisera against serotonin (*Figure 6C*) and GAD amplify morphological distinctions amongst these domains, such as delineating glomerular divisions in just the distal zone (dz in *Figure 6D*). Anti-DC0 immunocytology demonstrates expression of this antigen in all four zones, but at a much lower level than seen in mushroom body columns (*Figure 6E*). The same center has been identified at the same location and orientation, with respect to the optic lobes, in the shore crabs *Hemigrapsus nudus* and *H. oregonensis*, again revealing domains corresponding to those of the reniform body in *N. oerstedii*. A cluster of cell bodies provides a short pedestal composed of about two hundred smooth undecorated parallel processes that branch to four discrete neuropils (*Figure 6F,G*). However, in *H. nudus*, these and other discrete neuropils in the lateral protocerebrum either lack discernable DC0 or, if DC0 is expressed, it is at very low levels. An exception is a tract of axons leading from the optic lobes centrally (*Figure 6H*). Superficially, the reniform body is suggestive of a mushroom body-like center due to the arrangement of parallel processes that make up its pedestal. It is this feature that has lead to its erroneous identification as a mushroom body homologue (*Maza et al., 2016*). The rationale for rejecting that interpretation is provided in the Discussion.

## Discussion

### Mushroom bodies characterize insects and stomatopod crustaceans

It is broadly accepted that insects originated from crustaceans. Molecular phylogenies favor blind, morphologically simple anchialine crustaceans, called remipedes as the closest relatives of the insects (*Regier et al., 2005*; *Oakley et al., 2013*; *von Reumont et al., 2012*; *Schwentner et al., 2017*). This view conflicts with neural cladistics, which resolves insect brain organization closely corresponding to that of phylogenetically distant malacostracan crustaceans (e.g., Decapoda) (*Strausfeld and Andrew, 2011*). One source of conflict is that molecular phylogenies cannot provide reliable information about morphological changes that have occurred since related groups diverged. It is expected that through the course of evolution there will have arisen both genetic and structural novelties, as well as reversals and losses (*Chen et al., 2013*; *Jenner, 2004*), the latter impossible to code in relational trees based on morphological characters (*Strausfeld and Andrew, 2011*). For example, branchiopod crustaceans, closely related to both hexapods and remipedes, lack most features that define the brains of those two groups, and possess only a pair of nested optic neuropils (*Figure 5A*). Cephalocarids, a lineage closely related to Hexapoda and Remipedia, are also blind crustaceans, whose brains comprise predominantly olfactory centers and lack other cerebral neuropils common to hexapods and malacostracans (*Stegner and Richter, 2011*). Compared with Branchiopoda and Cephalocarida, the Remipedia midbrain shares more neural features with Malacostraca than it does with Hexapoda (*Fanenbruck et al., 2004*).

Similar cerebral arrangements in crustaceans and insects have frequently been ascribed to convergent evolution rather than to genealogical correspondence. This view (*Lozano-Fernandez et al., 2016*) can refer particularly to fundamental differences in insect and crustacean olfactory systems, even in crustaceans adapted to terrestrial life. In crustaceans, olfactory receptor neurons exclusively express ionotropic receptors, whereas in insects most olfactory receptor neurons are defined by their G-protein coupled receptors (*Stensmyr et al., 2005*; *Corey et al., 2013*; *Sato et al., 2008*). A neuropil ascribed exclusively to crustaceans is the dome-like second-order olfactory center called the hemiellipsoid body, which is supplied by the axons of thousands of projection neurons from numerous subunits of the olfactory lobe and its satellite neuropils (*Sullivan and Beltz, 2004*; *Sullivan and Beltz, 2005*; *Schmidt and Mellon, 2011*). Hemiellipsoid bodies also integrate olfactory, haptic and visual information (*Schmidt and Mellon, 2011*; *Mellon et al., 1992*).

A comparable arrangement pertains to insects, with a few notable differences. Fewer and smaller relay neurons, most restricted to single glomerular subunits of the antennal lobe (the primary olfactory lobe) send axons to second-order centers, the mushroom bodies (*Galizia and Rössler, 2010*). These inputs relay information about odorants, airborne molecules that are detected by the ligand-

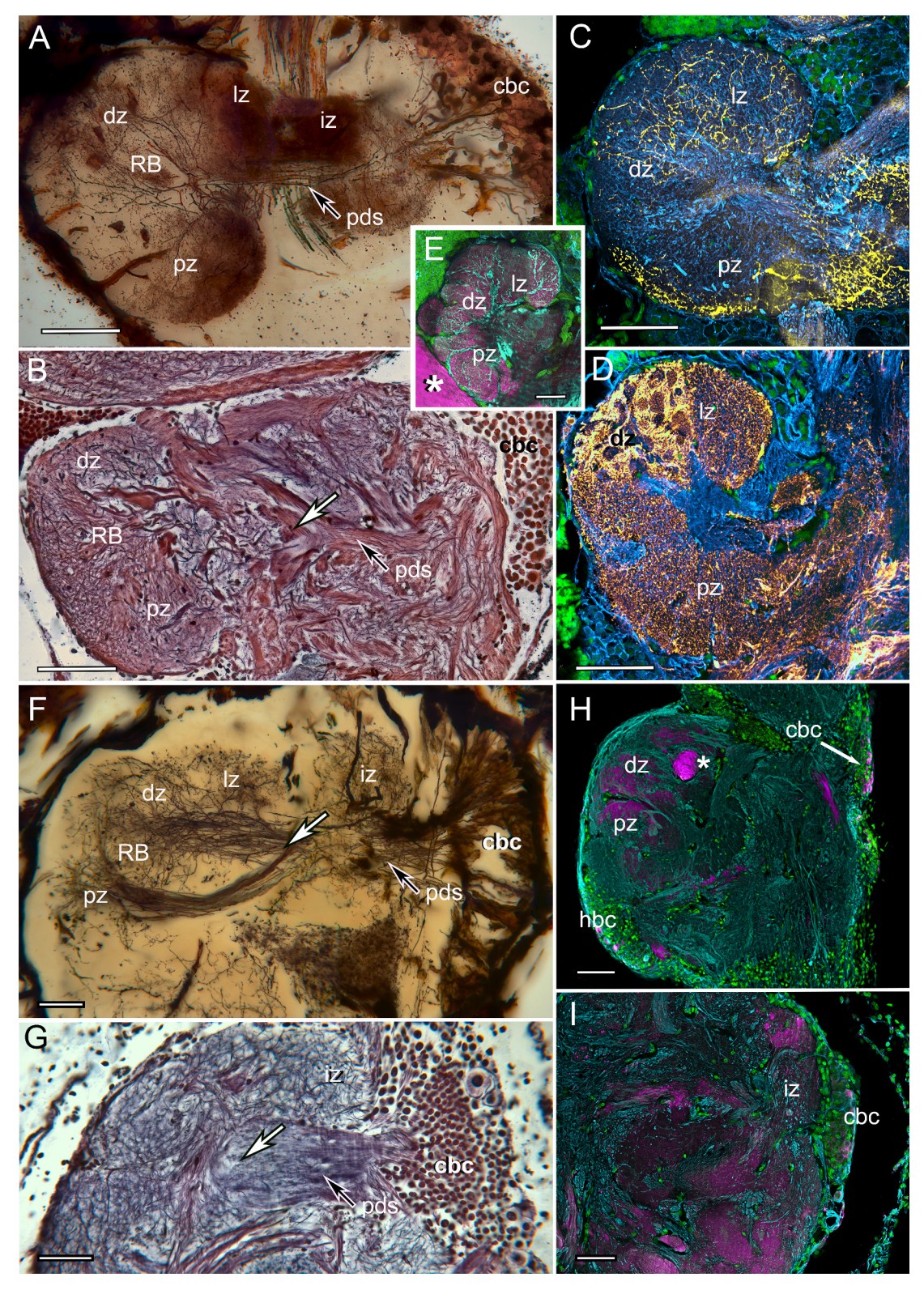

**Figure 6.** The reniform body: A multicomponent center possibly unique to Eumalacostraca. Neurohistological techniques confirm the presence of reniform bodies in the stomatopod *Neogonodactylus oerstedii* and the varunid shore crab *Hemigrapsus nudus*. (**A**) Golgi impregnation through a depth of 250 μm of the lateral protocerebrum of *N. oerstedii* showing that the reniform neuropils originate from a dense cluster of cell bodies (cbc) that provide a pedestal-like arrangement (pds) of smooth parallel neurites supplying four domains: an initial zone (iz), lateral zone (lz), distal zone (dz) and

*Figure 6 continued on next page*

*Figure 6 continued*

proximal zone (pz), which are also resolved in silver-stained thin sections (**B**). Antisera against serotonin (yellow) (**C**) and GAD (yellow) (**D**) further confirm distinctive zones in the stomatopod reniform body, the dz comprising discrete GAD-delineated glomerular divisions. (**E**) Anti-DC0 expressed in the lz, dz, and pz domains. Asterisk indicates part of a mushroom body column. (**F**) Golgi impregnation in *H. nudus*, showing three domains, lz, dz, and pz, corresponding to those in *N. oerstedii*. The cbc provides a pedestal-like domain comprising numerous smooth undecorated parallel neurites that bifurcate (at arrow) to provide the pz and dz+lz neuropils. This bifurcating trajectory is also resolved in silver-stained sections of *N. oerstedii* (arrow in *B*) and *H. nudus* (arrow in *G*). (**H, I**) DC0 (magenta) in *H. nudus* is absent or at very low levels except for a bundle of axons (asterisk in *H*) leading centrally from the optic lobes. Scattered neuron cell bodies in the cell body cluster (cbc) show elevated DC0 immunoreactivity (**H, I**), as do some cell bodies of the hemiellipsoid body (hbc). Scale bars: A-E, 100 μm; F-I, 50 μm.

DOI: https://doi.org/10.7554/eLife.29889.017

gated olfactory receptors. In many species, including *Drosophila*, mushroom bodies receive other modalities, including vision and haptic information (*Vogt et al., 2014*; *Paulk and Gronenberg, 2008*). However, because mushroom bodies are structurally so distinct from hemiellipsoid bodies, in that they are characterized by their extended lobes, they have been viewed as apomorphies of insects (*Stemme et al., 2016*; *Farris, 2013*; *Sandeman et al., 2016*).

To summarize, the combination of neuroanatomical characters that distinguish the insect mushroom body also distinguishes the stomatopod mushroom body (*Table 1*).

## The identity of the hemiellipsoid body

That hemiellipsoid bodies of only the closest eumalacostracan relatives of stomatopods possess some, but not all, mushroom body characters suggests that the hemiellipsoid body has derived from an ancestral mushroom body. Our analysis of five species of crown group Eumalacostraca show that only Alpheoidea, Stenopodidea (*Table 1*), and a single group of anomurans, the land hermit crabs, possess some mushroom body characters. There has been a reduction or complete loss of mushroom body columns in Anomura, Astacidea and Brachyura accompanied by an apparent elaboration of the ancestral calyx to provide a variety of morphologies of domed hemiellipsoid bodies, each characteristic of a genus (*Figure 5B*). A central debate has been whether hemiellipsoid bodies, which support multisensory functions comparable to those supported by insect mushroom bodies, can be viewed as their homologues or as crustacean apomorphies. The abundance of divergent hemiellipsoid body morphologies described by Sullivan and Beltz across an even broader range of taxa (*Sullivan and Beltz, 2004*; *Sullivan and Beltz, 2005*), and their observations of taxon-specific innervation by the olfactory tract, support an interpretation of the hemiellipsoid body as a highly divergent mushroom body calyx, a suggestion originally entertained by a remarkable study by Giuseppe Bellonci in 1882 (*Bellonci, 1882*).

## Visual habituation centers are not mushroom bodies

Reported activity in an elongated volume of neuropil situated close to the inner margin of the lobula of the varunid crab *Neohelice granulata* (*Maza et al., 2016*) suggested its identity as a learning and memory center. Recordings from this center showed long-term habituation in response to repetitive visual stimuli. It was proposed that this center is homologous to the insect mushroom body on the basis of a cluster of small cell bodies located ventro-medially in the eyestalk neuropil providing a columnar domain showing elevated p-CAMKII-α (*Maza et al., 2016*). Two territories populated by branched processes were referred to as vertical and medial lobes, adopting terms for insect mushroom body lobes. We have been unable to reconcile those neuroanatomical observations with characters defining the insect and stomatopod mushroom bodies. Parallel fibers comprising pedestal-like structures (*Figure 1E* in *Maza et al., 2016*) are shown as smooth, lacking spines and varicosities. They also lack efferent and afferent domains that would provide orthogonal networks typical of mushroom bodies. Dendrites described as clawed (*Maza et al., 2016*, *Figure 1F*) appear to be afferent terminals. Immunoreactivity to anti-pCaMKII shows elevated expression tightly constrained to the region of cell body neurites, after which its levels diminish in unspecific neuropil. In all respects, the described neuropil in *N. granulata* corresponds to the *Hemigrapsus* reniform body (*Figure 6F–I*). Although more elaborate in Stomatopoda (*Figure 6A–E*), its reniform body neuropils unambiguously correspond to those attributed to the *N. granulata* habituation center and to *H. nudus*. We are unable to show evidence for any lobed center that meets the relevant criteria for a mushroom body

in varunids and suggest that the reniform body is an intriguing visual association center that may be unique to eumalacostracan crustaceans. Nothing comparable has yet been identified in the *Drosophila* lateral protocerebrum or in that of any other insect.

## Are mushroom bodies a key indicator of pancrustacean ancestry?

Our survey across Pancrustacea (Crustacea+Hexapoda) identifies mushroom bodies in stomatopods and insects. No other eumalacostracan crustacean outside Stomatopoda has yet provided clear evidence of an insect-like mushroom body.

Comparisons across this pancrustacean phylogeny (*Figure 5A*) thus suggest two interpretations. One is that mushroom bodies evolved convergently in insects and stomatopods. The other is that the mushroom body is an ancestral attribute of the pancrustacean brain that has been retained in just the hexapod and stomatopod lineages but modified or lost in all others. Fossil brains described from the lower Cambrian stem arthropod *Fuxianhuia protensa* suggest that optic lobe and neuromeric arrangements of the brain of this ancient stem arthropod typify those of extant malacostracans and insects: the presence of three nested visual centers arising from a cerebrum comprising a fused forebrain, midbrain and hindbrain, each neuromere denoted by its relationships with the compound eyes, antennules and paired post-antennular appendages (*Figure 5A*, *Figure 5—figure supplement 3A*) (*Ma et al., 2015*; *Ma et al., 2012*). Although fossils cannot reveal the organization of centers within the brain, those preserved features described above are cardinal indicators of an ancestral ground pattern that originated more than half a billion years ago at the onset of the lower Cambrian. Subsequent lineages originating at an estimated time of 487 million years ago provide today's crown Pancrustacea (*Collette and Hagadorn, 2010*). Taking neuropils that have been identified as common to Pancrustacea and superimposing these onto the ancestral ground pattern (*Figure 5—figure supplement 3B*) suggests a plausible ancestral arrangement in such a brain (*Figure 5—figure supplement 3C,D*). That mushroom bodies have been demonstrated in Myriapoda, the sister group of Pancrustacea, and Chelicerata the sister group to all mandibulate arthropods, further supports a very ancient origin of this center (*Wolff and Strausfeld, 2016*).

Attendant to such considerations is that many elements of the ancestral ground pattern have been lost or drastically reduced in numerous crustacean lineages, even to just a few neurons such as in Cirripedia and Cephalocarida (*Callaway and Stuart, 1999*). Simplification from genealogically ancient complexity is common in nature at many phylogenetic levels (*Collin et al., 2009*). It should come as no surprise to find examples of simplification or independent retentions of past complexity in the neuromorphological landscape.

## Are insect and stomatopod mushroom bodies homologous or convergent?

The demonstration that phenotypically identical organs in widely separated species can be ascribed to convergent genomic evolution (*Pankey et al., 2014*) suggests caution in claiming homology of identical phenotypes. Thus, although corresponding morphologies in stomatopods and insects suggest mushroom body homology, transcriptomic evaluation must, eventually, be the arbiter of whether these centers indeed correspond genealogically. At present, the strongest argument against is that the sister group of all hexapods (Remipedia) and the sister group of all Malacostracans (Leptostraca) possess simple hemiellipsoid bodies, not mushroom bodies (*Fanenbruck et al., 2004*; *Kenning et al., 2013*).

Convergent evolution would imply that mushroom bodies in insects and stomatopods evolved independently in response to comparable selective constraints. The most obvious would be the appearance of biotopes requiring superior vision. Stomatopods and certain insects possess superior optics and underlying optic lobe circuitry that enable the detection and discrimination of visual flow fields, chromatic features, patterns and structures. However, speaking against this are archaeognathan insects, the sister group of all other insects (*Misof et al., 2014*), which have a more elaborate visual system than those of apterygote Zygentoma, the sister group of all winged insects (*Misof et al., 2014*). Yet archaeognathans lack mushroom bodies. *Lepisma saccharina*, which is a typical zygentoman insect, has diminutive eyes and reduced optic lobes but possesses robust mushroom bodies equipped with calyces and elaborate lobes (*Farris, 2005*). Amongst crustaceans, many eumalacostracan species possess prominent eyes surmounting elaborate visual centers. Shore crabs,

for example, have excellent vision and highly elaborated optic lobes (*Sztarker et al., 2005*; *Berón de Astrada et al., 2001*), but they lack mushroom bodies. Given these contradictions, what aspects of behavior specific to stomatopods and insects might be relevant to mushroom body evolution? One proposed driver of the evolution of large mushroom bodies is the requirement to recall the exact locations and properties of places from which to obtain nourishment. Heliconid butterflies maintain their enlarged mushroom body lobes when allowed repeated visits to distributed foraging sites (*Montgomery et al., 2016*). The large mushroom bodies of social hymenopterans are suggested to have evolved in conjunction with the evolution of parasitic behavior, where individuals learn the many locations of potential hosts (*Farris and Schulmeister, 2011*). It may be significant, therefore, that the only eumalacostracan groups in addition to stomatopods that evidence memory of exact locations are cleaner shrimps, pistol shrimps, and land hermit crabs, all of which have mushroom body-like attributes (*Figure 5B – Figure 5—figure supplement 1B–D*, *2A*). Cleaner shrimps are renowned for their fidelity to specific locations visited by the fish they clean (*Limbaugh et al., 1961*). Pistol shrimps are the only crustaceans to have evolved eusociality (*Duffy, 1996*), an attribute requiring memory of position both outside and within the community. And land hermit crabs are known for their navigational skills and memory of sites at which they socially interact (*Rotjan et al., 2010*). There are, then, valid grounds to suppose convergent evolution of mushroom bodies in certain crustaceans and hexapods.

Stomatopod and insect mushroom bodies also pass three crucial tests for phenotypic homology (*Patterson, 1988*). These are their similarity, the exclusion of competing structures that might suggest a similar interpretation (conjunction), and the co-existence of additional homologies (congruence). The present results have demonstrated structural identicalities. Reniform bodies, proposed as a homologue of the insect mushroom body (*Maza et al., 2016*), are excluded because in stomatopods they coexist with mushroom bodies. The third condition, that structures in two species are more likely to be homologues if they share other homologies, is supported by the locations of other corresponding brain centers in insects and stomatopods (*Figure 1—figure supplement 2*). Specific examples are the fan-shaped central complex neuropils in the midbrain of stomatopods and insects connected to identical suites of satellite neuropils (*Thoen et al., 2017*).

We are obliged to conclude that there is, as yet, no definitive conclusion. Phenotypic and genomic homology would support the hypothesis that mushroom bodies evolved very early in pancrustacean or even in panarthropod evolution (*Wolff and Strausfeld, 2016*). That identical centers of such stunning complexity may have evolved convergently in stomatopods and insects is just as fascinating and should lead to a greater understanding of the overarching significance of mushroom bodies in arthropod behavior.

## Materials and methods

### Animals

One hundred and eight mature stomatopods (both sexes), *Neogonodactylus oerstedii*, were obtained commercially from waters off the coast of Florida. Forty-seven *Gonodactylus smithii* were obtained from designated areas overseen by the Lizard Island Research Station, Australia (GBRMPA Permit no. G12/35005.1, Fisheries Act no. 140763). Five *Stenopus hispidus* (Caribbean) and 25 *Alpheus bellulus* (Caribbean) were obtained from LiveAquaria.com, Rhinelander, WI, USA. Animals were maintained isolated in small perforated transparent 'Ziplock' containers immersed in running artificial seawater. *Drosophila melanogaster* and *Periplaneta americana* were reared in laboratory cultures at the University of Arizona, Tucson and Washington University, Seattle. Five specimens of *Coenobita clypeatus* were purchased locally and maintained in a vivarium. Thirty *Hemigrapsus nudus and H. oregonensis* were collected from designated sites on San Juan Island, WA, USA. Six *Procambarus clarkii* were purchased from domestic suppliers (Carolina Biological Supply Company, Burlington, NC). Golgi impregnations were performed on 72 *N. oerstedii* and 40 *G. smithii*. Each immunocytological method was performed identically on at least 3 individuals. Bodian and Golgi silver staining was performed on 6 *N. oerstedii*. Cell body counts were obtained from two mushrooms bodies from *N. oerstedii* and *P. americana*.

## Antibodies and immunocytology

A monoclonal antiserum against α-tubulin (DSHB Cat# 12G10 anti-alpha-tubulin, RRID:AB_1157911) used at a concentration of 1:100, was obtained from the Developmental Studies Hybridoma Bank developed under the auspices of the NICHD and maintained by the University of Iowa, Department of Biology (Iowa City, IA). Monoclonal antiserum against synapsin (anti-SYNORF1, DSHB Cat# 3C11RRID:AB_528479) obtained from the same source were used at a concentration of 1:100. Polyclonal antiserum raised against serotonin (ImmunoStar, Hudson, WI, Cat# 22941, RRID:AB_572268) was used at a concentration of 1:500. Anti-DC0 (D. Kalderon, Columbia University; New York; USA Cat# DC0, RRID:AB_2314291), a generous gift from Dr. Daniel Kalderon, was used at a concentration of 1:250 for immunohistochemistry. Anti-Leonardo, a generous gift from Dr. Ronald Davis was used at a concentration of 1:500 and anti-pCaMKII (Santa Cruz Biotechnology Cat# sc-12886-R, RRID:AB_2067915) was used at a concentration of 1:100 for immunohistochemistry. Each antiserum was from aliquots demonstrated for specific affinity to *Drosophila* mushroom body lobes. As pioneered by studies on remipede brains (*Stemme et al., 2016*), a polyclonal antibody raised against of the synthesizing enzyme glutamic acid decarboxylase (GAD) (Sigma-Aldrich Cat# G5163, RRID:AB_477019) was used to detect putative GABAergic neurons. A monoclonal antibody against tyrosine hydroxylase (TH) raised in rats (ImmunoStar, Hudson, WI Cat# 22941, RRID:AB_572268) which had been previously shown to specifically label dopaminergic neurons in the lobes of insect mushroom bodies was used on intact stomatopod and cockroach brains.

In preparation for immunocytology, animals were immobilized by refrigeration at 4°C and the brains were dissected free in cold (4°C) fixative containing 4% paraformaldehyde in PBS, pH 7.4 (PBS from Sigma, St. Louis, MO). 10% sucrose was added to the fixative solution for marine organisms. Holes were cut in the retina and in cuticle covering eyestalk neuropils. The eyestalks were then severed and immersed in fixative and placed in a microwave adjusted to 18°C for two cycles of 2 min with power, and then 2 min under vacuum. Next, the brains were left in fresh fixative overnight at 4°C (but see below for TH immunostaining). The following day brains were washed three times, 10 min each, in PBS and then embedded in albumin gelatin. Embedded tissue was cut into 60 μm serial sections using a vibratome (Leica, Nussloch, Germany). Individual sections were placed in wells of a well plate (1 ml capacity) for further processing. Sections were washed six times over 20 min in PBS containing 0.5% Triton X-100 (Electron Microscopy Supply, Fort Washington, PA, Cat. no. 22140; PBS-TX). Then 50 μL normal serum (Jackson ImmunoResearch Labs Cat# 017-000-121, RRID:AB_2337258) was added to each well containing 1000 μL PBS-TX. After 1 hr, primary antibody or antiserum was added to each well and the well plate was left on a slow agitating shaker overnight at room temperature. The next day, sections were washed six times over 3 hr in PBS-TX. Then 1000 μL aliquots of PBS-TX were placed in tubes with 2.5 μL of secondary Cy2, Cy3, or Cy5 conjugated IgGs (Jackson ImmunoResearch, West Grove, PA, respectively Cat# 715-225-150, RRID:AB_2340826, 715-165-150, RRID:AB_2340813, and 715-175-150, RRID:AB_2340819) and centrifuged at 13,000 rpm for 15 min at 4°C. A 900 μL aliquot of this solution was added to each well. The well plate was left on a shaker to gently agitate the sections overnight at room temperature. Tissue sections were next washed six times in PBS over 3 hr, embedded on glass slides in a medium of 25% polyvinyl alcohol, 25% glycerol and 50% PBS, for imaging using confocal microscopy. Where applicable, sections were rinsed in 0.01M tris-HCl buffer (pH 7.5; Sigma, T1503) and incubated in the fluorescent nuclear stain Syto-13 (Life Technologies, Grand Island, NY) at a concentration of 1:2000 prior to mounting on glass slides.

For labeling F-actin, preparations were left to incubate in phalloidin conjugated to Alexa Fluor 546 (Thermo Fisher Scientific Cat# A-22283, RRID:AB_2632953) following secondary antibody incubation at a concentration of 1:40 in 0.5% PBST for three days at room temperature on a gentle shake.

For tyrosine hydroxylase (TH) immunocytology, eyestalks were fixed at 4°C for 30 min in 0.01M phosphate buffered 4% paraformaldehyde containing 10% sucrose. The tissue was then dissected free, and whole eyestalk neuropils were soaked for 2 hr in PBS containing 1% Triton X-100, before being transferred to a vial containing 1% PBS-TX and 5% normal donkey serum (Jackson ImmunoResearch Labs Cat# 017-000-121, RRID:AB_2337258). After 2 hr, primary antibody diluted 1:250 was added to the container, which was left on a slow agitating shaker for 48 hr at room temperature. Tissue was then washed several times in 0.5% PBS-TX over 3 hr before treatment with Cy5 conjugated

IgGs, as above. Tests using an antiserum against dopamine (Polyclonal Rabbit anti Dopamine (Abcam Cat# ab8888, RRID:AB_306841) resolved identical neurons but with poor resolution of fine processes and twigs, hence the exclusive use of TH for this study. Secondary antibody treatment lasted for 24 hr followed by several washes in PBS-TX, cleared in glycerol and then mounting in welled glass slides under glycerol for confocal microscopy. Tissue was subjected to the two cycle microwave treatment described above once every 24 hr while the tissue was incubating in either primary or secondary antibody. When labeling tissue with a second primary antibody, such a serotonin, in conjunction with TH, after whole tissue labeling the specimen was again fixed as if it were fresh, for an additional 12 hr at 4°C in 4% paraformaldehyde in PBS. The next day, the tissue was embedded and sectioned as described above, and subjected to the relevant immunohistochemical treatment.

## Reduced silver staining

Bodian's original (*Bodian, 1936*) method was used on brains fixed in AAF (16 ml 80% ethanol, 1 ml glacial acetic acid, 3 ml 37% formaldehyde), dehydrated, cleared in terpineol, and embedded in Paraplast Plus (Sherwood Medical, St. Louis, MO).

## Golgi impregnations

Stomatopods were briefly immobilized by covering with granulated ice. The head and eyestalks were removed and neural tissue dissected and desheathed in 1 part 25% glutaraldehyde and 5 parts 2.5% potassium dichromate adjusted with sucrose (*Thoen et al., 2017*). After a suitable incubation period, tissue was washed in potassium dichromate before immersion in a 0.01% osmium tetroxide carried in 2.5% potassium dichromate for 12 hr before a 24 hr immersion in 0.75% silver nitrate. These last two steps were then repeated before embedding tissue in Durcupan plastic and serial sectioning at 40 µm. Images through depths of 50–100 µm were obtained using serial step-through focusing and reconstitution using Adobe Photoshop.

## Imaging and reconstruction

Immunohistological images were collected using a Zeiss Pascal or 880 confocal microscope. Light microscopy images were obtained with a Leitz Orthoplan microscope. Step focus series of stitched images were reconstructed using the software Helicon Focus (Helicon Soft, Kharkov, Ukraine; RRID: SCR_014462). Three-dimensional reconstruction of the hemiellipsoid body and its lobes employed 220 stacked 12 µm-thick serial silver-stained and interpolated profiles processed using ImageJ (RRID:SCR_003070). Sections were prealigned with reference to 5 fiducial landmarks provided by non-neural elements, such as blood vessels and neural sheath extending vertically through the section series.

## Globuli cell densities

Bodian-stained brains of *N. oerstedii* and *P. americana* were used for estimating densities of globuli cells, which provide mushroom body parallel fibers. Serial photographs at 1 µm intervals through a depth of 10 µm captured images of perikarya, their nuclei and nucleoli. Counts of the latter were made within $50 \times 50 \times 10$ (25,000 µm³) volumes. Averages were calculated for globuli cell packing densities within 1 mm³, arriving at $3.48 \times 10^6$/mm³ for *N. oerstedii* and $4.8 \times 10^6$/mm³ for *P. americana*. The total volume of the globuli cell population was calculated from serial reduced-silver sections. For *N. oerstedii* images were taken at $1173 \times 846$ pixels at a resolution of 0.5068 pixels per micron. Voxel size was calculated as $1.9730 \times 1.9730$ µm³, with 12-µm-thick sections, resulting in 6.08 pixels per z-level. *P. americana* was imaged at $2080 \times 1542$ pixels at a resolution of 2.02 pixels per micron. Voxel size was calculated as 0.4950 µm³ 12-µm-thick sections resulting in 24.24 pixels per z-level. Sections were aligned as described above and the regions containing the globuli cell population were segmented using the TrakEM2 plugin in Fiji (*Cardona et al., 2012*; *Schindelin et al., 2012*) using the TrakEM2 measuring tool to calculate the total volume.

### Reconstruction of the lower Cambrian stem arthropod brain

Depiction of the fossilized brain of *Fuxianhuia protensa* (Figure S6) was obtained by superimposing the imaged brain of specimen YKLP 11357 onto the montaged part and counterpart of YKLP 15006. Original data for these specimens is published in *Ma et al., 2015*.

### Terminology

Terms and abbreviations follow the recommendations of the insect brain consortium that has devised terms applicable to both insect and crustacean (Pancrustacea) central nervous systems. These are provided and comprehensively explained in *Ito et al. (2014)*.

## Acknowledgements

This research was supported by a grant to NJS from the US. Air Force Research Laboratory (FA8651-13-1-0001), the University of Arizona's Center for Insect Science, and funding from the University of Arizona Regents' Fund. Support for HHT and JM came from grants awarded by the Asian Office of Aerospace Research and Development (AOARD- 12–4063) and the Australian Research Council (FL140100197), and a Doctoral Fellowship (2013) to HHT from the Lizard Island Research Foundation and the Lizard Island Research Station, a facility of the Australian Museum. We thank Roy Caldwell, University of California, San Diego for providing the image for *Figure 1A*. We thank Camilla Strausfeld for critically discussing and editing the text. We thank Todd Oakley (University of Santa Barbara, California) for advice and suggestions regarding genetic convergence and Michael Sanderson (University of Arizona) for insights regarding character assignation. We have also have profited from exchanges with Gerhard Scholtz (Humboldt University, Berlin), Greg Edgecombe (Natural History Museum, London), and Frank Hirth, Kings College, London

## Additional information

### Funding

| Funder | Grant reference number | Author |
| --- | --- | --- |
| Air Force Research Laboratory | FA8651-13-1-0001 | Nicholas James Strausfeld |
| Asian Office of Aerospace Research and Development | AOARD- 12-4063 | Justin Marshall |
| Australian Research Council | FL140100197 | Justin Marshall |
| Australian Research Council | Doctoral Fellowship 2013 | Hanna Halkinrud Thoen |

The funders had no role in study design, data collection and interpretation, or the decision to submit the work for publication.

### Author contributions

Gabriella Hannah Wolff, Conceptualization, Data curation, Investigation, Visualization, Methodology, Writing—original draft, Project administration, Writing—review and editing; Hanne Halkinrud Thoen, Conceptualization, Formal analysis, Investigation, Visualization, Methodology, Writing—original draft, Writing—review and editing; Justin Marshall, Conceptualization, Formal analysis, Funding acquisition, Investigation, Visualization, Methodology, Writing—original draft, Project administration, Writing—review and editing; Marcel E Sayre, Investigation, Visualization, Methodology; Nicholas James Strausfeld, Conceptualization, Data curation, Project administration, Funding acquisition, Investigation, Visualization, Methodology, Writing—original draft, Writing—review and editing

### Author ORCIDs

Gabriella Hannah Wolff, https://orcid.org/0000-0002-0075-4975
Nicholas James Strausfeld, https://orcid.org/0000-0002-1115-1774

### Decision letter and Author response

Decision letter https://doi.org/10.7554/eLife.29889.019

Author response https://doi.org/10.7554/eLife.29889.020

## Additional files

**Supplementary files**
• Transparent reporting form
DOI: https://doi.org/10.7554/eLife.29889.018

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
