## [Decision Letter]

Thank you for submitting your article "An insect mushroom body in a crustacean brain" for consideration by *eLife*. Your article has been reviewed by three peer reviewers, and the evaluation has been overseen by a Reviewing Editor and Eve Marder as the Senior Editor. The following individual involved in review of your submission has agreed to reveal his identity: Kei Ito (Reviewer #1).

The reviewers have discussed the reviews with one another and the Reviewing Editor has drafted this decision to help you prepare a revised submission. There were two critical issues identified that need to be resolved: (23) The justification for the 15 criteria of MB identification should be improved. Each criterion should be independent from others. (67) The authors should provide a more convincing argument that the observed similarity is homology instead of convergence. i.e., why can't this be just convergence?

The comments of the three reviewers are provided below more or less in entirety because we think it is important for you to see "where they are coming from" as they phrase their positions. Note that they are not asking for new experimental work, but are challenging some of the ways you are analyzing and interpreting the findings. We ask that you revised the manuscript to address the major concerns and look forward to receiving a revision.

Reviewer #1:

In this study the authors employed an approach to characterize the feature of the mushroom body with 15 morphological and biochemical criteria, and identified a specific neuronal structure in the lateral protocerebrum of the mantis shrimp Stomatopoda brain that is likely to correspond to the insect mushroom body. This approach has been proven to be a powerful tool by their previous publications for identifying correspondences between the brains of distant species while excluding superficial resemblance. Past studies did not identify any structure in crustacean brains that would truly correspond to the insect mushroom body, making the evolutionary comparison difficult across Pancrustacea. The Stomatopoda brain has been shown to share many characteristics with the Hexapoda brain, and the current finding fills the final missing part. The authors further compared the mushroom body-like structures in other Eumalacostraca brains, and deduced a hypothesis that the brains of Hexapoda and Stomatopoda should retain the most ancient and complex architecture of the Pancrustacean ancestral ground pattern, whereas the brains of other species can be explained by selective partial losses or modifications from this ground pattern.

Those results are highly important not only for the people who study comparative neurology but also for a broad audience who wants to understand the evolution of brain architecture since the Cambrian period. The images of the brain samples are clear and beautiful, and schematic drawings are easy to understand. Because of its high quality analysis and high impact on the understanding of the brain evolution, the study should deserve publication in a journal like *eLife*.

I have a couple of relatively major concerns/questions that the authors should be able to address:

1) Unlike more specialized journals, ordinary readers of *eLife* are unfortunately not familiar with the classification system of arthropods. Extensive appearance of the names of the Classes, Subclasses, Orders and Suborders would alienate potential readers such as molecular geneticists who would have keen interests in brain evolution but only limited knowledge in taxonomy. Addition of the plain English names for the terms is helpful, but it alone may not be enough, because those people who mostly work with test tubes and restriction enzymes are unfamiliar even with such plain names. The message of this study should be highly intriguing to such people, though. Schematic illustrations of organisms in the bottom panel of Figure 5 are very intuitive for such people. It would be very helpful, if the authors could also provide such drawings for other taxonomical terms that appear in the text (such as those in the top-right panel of Figure 5).

2) One of the characteristic features of the insect mushroom bodies with calyces is microglomeruli. There, dendrites of a small number of intrinsic neurons (globuli cells) contact with a presynaptic bouton of an olfactory projection neuron. Sparse coding is achieved because a globuli cell sends dendrites only to a limited number of microglomeruli. Could this be another criterion to characterize the mushroom body? Or, is it too specific to be true in only certain species?

3) Because the Stomatopoda mushroom body is situated so close to the optic lobe, and because hemiellipsoid bodies are known to integrate olfactory and visual signals, does the Stomatopoda mushroom body receive input also from the visual system? If so, does it supply the calyces, or lobes? Would it be worthwhile to compare also with the Hymenopteran brain with massive visual inputs?

4) It is not clear why only Hexapoda and Stomatopoda retained the ancestral ground plan while partial or massive structural loss and alteration occurred in other branches. It would be difficult to provide conclusive evidences at this stage of the study, but it would be helpful, if the authors could suggest possible mechanisms or principles. Is there any systematic correlation between the degree/tendency of structural alteration and the behavioral/functional complexity of the organisms in the respective branches?

Reviewer #2:

This manuscript describes novel and interesting features of stomatopod higher brain centers. Stomatopods have highly developed visual systems and visual behaviors that may be associated with the extreme complexity of higher brain centers (hemiellipsoid bodies, which I abbreviate as HEB). The neuroanatomical data and interpretations are outstanding.

However, the comparative data presented in this manuscript and available in the literature do not support the authors' primary conclusion: that the similarities between the stomatopod HEBs and insect MBs reflect an ancestral feature of the pancrustacean brain. The authors' attempts to pigeonhole the data into a case for homology creates a confusing quagmire surrounding an otherwise solid account of extreme structural variation in malacostracan HEBs.

Beginning in the Introduction and Table 1, the authors flip-flop repeatedly in their assessment of features of the HEB in the studied species. The difficulty in understanding how HEBs of malacostracans differ is compounded by the questionable rationale and utility of the characteristics chosen to score MB-like structures (Table 1).

On the first point: this group has published several studies on the land hermit crab *Coenobita clypeatus*, belonging to the eumalacostracan Infraorder Anomura (Brown and Wolff, 2012; Wolff et al., 2012 and Wolff and Strausfeld, 2015). Wolff et al., 2012 states that "Comparisons of the morphology, ultrastructure, and immunoreactivity of the hemiellipsoid body of *C. clypeatus* and the mushroom body of the cockroach *P. americana* reveal in both a layered motif provided by rectilinear arrangements of extrinsic and intrinsic neurons as well as a microglomerular organization" and "…the superior part (of the HEBs) approximates a mushroom body's calyx in having large numbers of microglomeruli." These papers clearly suggest that the HEBs of *C. clypeatus* are similar in many regards to insect MBs.

However, the authors flip flop on the similarity between *C. clypeatus* HEBs and insect MBs multiple times in the text.

Sentence from the Introduction section:.…"although crustaceans have paired higher centers in their lateral protocerebra, those centers share none of the identifying characteristic(s) of lobed mushroom bodies as described above."

*C. clypeatus* HEBs have many features in common with insect MB, including orthogonal arrangements of inputs and outputs, but then later in the Introduction the authors state that *C. clypeatus* HEBs lack orthogonal circuitry.

In Table 1, Anomura (which includes *C. clypeatus*) is scored as lacking a "domed center or calyces" "lobes intersected by orthogonal inputs," and "lobes partitioned as sequential domains." Table 1 thus directly contradicts the findings of Wolff et al., 2012.

On the second point, the data and analysis used to generate Table 1. The traits selected to determine the identity of mushroom bodies are very confusing and their selection is not justified in the text.

1) Some traits are redundant (what is the difference between cell bodies supplying the lobes and supplying the calyx? The same cell population supplies both.)

2) The absence of some traits exclude others (lack of a calyx necessarily means the animal lacks cell bodies supplying the lobes, clawed distal dendrites, calyx contains orthogonal networks). This leads to differences between species that appear more dramatic than they actually are.

3) Calyx/dome are grouped together in some traits, while Lobes/dome are grouped in others. Is the HEB dome considered analogous/homologous with the MB calyces or lobes?

4) Some just don't make sense. I was under the impression that orthogonal networks are characteristic of MB lobes, not calyces.

5) The characters do not take into account any aspect of behavior or sensory ecology of the animal. For example, anosmic insects would score 11/15 traits. If the MBs of insects and stomatopods are presumed to be basal, are we to assume then that *Drosophila* is basal to cicadas and backswimmers?

6) The authors do not present data to back up scoring for some traits in some species. The Verunidae are scored as lacking "lobes or dome resolved with anti-DC0." However, Figure 5—figure supplement 2 shows the verunid *Hemigrapsis nudus* with DC0 labeling in the HEB. The caption for Figure 5 states that "Mushroom body lobes are recognized by their DC0 immunoreactivity across pancrustaceans." Verunids are also described as lacking orthogonal networks, but I don't understand how it's possible to determine this without Golgi impregnations to resolve individual neurons.

7) There are so many "no data" entries for most of the species, primarily due to lack of literature, that it is inevitable that stomatopods and insects will appear to share the most characters.

In conclusion, this reviewer feels that factors described in the literature and discussed by the authors strongly support not homology, but convergence of the stomatopod HEBs and insect MBs. HEBs are found in a number of crustacean taxa, including some outside of the Malacostraca. In the few species studied in detail such as *C. clypeatus*, the HEBs share many characteristics with insect MBs, providing a basis for proposing that the complex HEBs of stomatopods through modification of HEBs. That HEBs gave rise to MB-like HEBs in one taxon of Crustacea, as simpler MBs gave rise to more complex MBs in some insects as a result of similar selective pressures for visual behavior is a far more parsimonius interpretation of the data presented and the existing literature. The rationale provided by the authors for homology is inconsistent and unconvincing.

Reviewer #3:

In their manuscript "An insect mushroom body in a crustacean brain" Gabriella H. Wolff and co-authors give an in-depth neuroanatomical and immunocytochemical analysis of a prominent neuropil in the lateral protocerebrum of mantis shrimps (Stomatopoda), that serves as second stage of the central olfactory pathway and traditionally has been known as hemiellipsoid body (HB). With this analysis the authors provide substantial evidence supporting the notion that this neuropil is in fact homologous to the well-known mushroom body (MB) of the insect brain. By additionally analyzing the expression of a protein required for learning and memory in *Drosophila* (DC0) in the HB in some other crustacean taxa, the authors arrive at a detailed reconstruction of the phylogenetic development of the HB/MB within Pancrustacea, the monophyletic taxon comprised of Crustacea and Hexapoda. Furthermore, the authors provide a less-detailed neuroanatomical description of a hitherto unknown neuropil in the lateral protocerebrum of crustaceans, the 'reniform body' – which in some neuroanatomical aspects is similar to the HB/MB.

Overall, the study is very well executed and the main results are presented in detailed and very informative reconstruction drawings as well as in numerous images of original neuroanatomical and immunocytochemical data.

There is, however, one main conceptual issue that the authors should address as well as numerous minor suggestions and comments. The main conceptual issue on which the paper hinges, is what constitutes a MB and distinguishes it from a HB – and the authors should improve their definition(s) substantially. First, there is some contradiction within the paper itself that needs to be reconciled: In Results paragraph two the authors state that a MB is identified by 12 of the 15 criteria provided in Table 1 – but they fail to identify this set of 12 criteria (the 'HBs' of stomatopods correspond to the MBs of insects in a 15 criteria according to Table 1). Furthermore, later in the Results (subsection “Divergent evolution of the mushroom bodies”) the authors talk about 'mushroom body lobes' within HBs (which is a truly confusing description and concept) that apparently are identified solely by the expression of DC0. Even more problematic is the lack of a definition of what constitutes a 'lobed center' in the lateral protocerebrum: This is the first criterion given in Table 1 that distinguishes MBs from other neuropils in the lateral protocerebrum, which at least for Varunidae according to the authors' data include the 'reniform body' that they describe as a 'lobate center'. So why then is that not a lobed center? In fact, the term 'lobe' may be one of the least distinct anatomical terms and without a crystal-clear definition it seems impossible to exclude neuropil structures of typical HBs (or other neuropils in the lateral protocerebrum) by it. Most of the other criteria in Table 1 of the remaining 14) depend on the presence of a 'lobe' and therefore do not provide additional independent characters differentiating between HBs and MBs (it is odd that for Anomura which do not have lobed centers the density of cell bodies supplying lobes is checked). The terms 'dome' and 'calyx' also prominently appear in the Table – and they need to be rigorously defined to be useful.

---

## [Author Response]

The reviewers have discussed the reviews with one another and the Reviewing Editor has drafted this decision to help you prepare a revised submission. There were two critical issues identified that need to be resolved: (23) The justification for the 15 criteria of MB identification should be improved. Each criterion should be independent from others. (2) The authors should provide a more convincing argument that the observed similarity is homology instead of convergence. i.e., why can't this be just convergence?The comments of the three reviewers are provided below more or less in entirety because we think it is important for you to see "where they are coming from" as they phrase their positions. Note that they are not asking for new experimental work, but are challenging some of the ways you are analyzing and interpreting the findings. We ask that you revised the manuscript to address the major concerns and look forward to receiving a revision.

We have greatly appreciated these detailed reviews, which are constructive and fair. We have extensively revised the manuscript with the aim of clarifying the text such that the description of and rationale for the characters is now lucid and unambiguous. Although you requested no additional data, in response to questions of two of the reviewers we have obtained evidence for microglomeruli, synaptic complexes typifying the insect mushroom body calyx, which we have now identified in the stomatopod mushroom body as well. We have added a balanced discussion debating evidence for the two alternative interpretations of the stomatopod mushroom body: its derivation from an ancestral organization also shared by insects, but lost in all other pancrustaceans; or, that it is a fascinating example of convergent evolution driven by specific ecological/behavioral constraints. Although evidence supports phenotypic homology, there can be no resolution to these alternatives until the transcriptomic underpinnings of both the stomatopod and insect mushroom bodies have been resolved. That long-term research program planned with the Oakley laboratory will be founded on the present account.

Reviewer #1:In this study the authors employed an approach to characterize the feature of the mushroom body with 15 morphological and biochemical criteria, and identified a specific neuronal structure in the lateral protocerebrum of the mantis shrimp Stomatopoda brain that is likely to correspond to the insect mushroom body. This approach has been proven to be a powerful tool by their previous publications for identifying correspondences between the brains of distant species while excluding superficial resemblance. Past studies did not identify any structure in crustacean brains that would truly correspond to the insect mushroom body, making the evolutionary comparison difficult across Pancrustacea. The Stomatopoda brain has been shown to share many characteristics with the Hexapoda brain, and the current finding fills the final missing part. The authors further compared the mushroom body-like structures in other Eumalacostraca brains, and deduced a hypothesis that the brains of Hexapoda and Stomatopoda should retain the most ancient and complex architecture of the Pancrustacean ancestral ground pattern, whereas the brains of other species can be explained by selective partial losses or modifications from this ground pattern.Those results are highly important not only for the people who study comparative neurology but also for a broad audience who wants to understand the evolution of brain architecture since the Cambrian period. The images of the brain samples are clear and beautiful, and schematic drawings are easy to understand. Because of its high quality analysis and high impact on the understanding of the brain evolution, the study should deserve publication in a journal like eLife.I have a couple of relatively major concerns/questions that the authors should be able to address:1) Unlike more specialized journals, ordinary readers of eLife are unfortunately not familiar with the classification system of arthropods. Extensive appearance of the names of the Classes, Subclasses, Orders and Suborders would alienate potential readers such as molecular geneticists who would have keen interests in brain evolution but only limited knowledge in taxonomy. Addition of the plain English names for the terms is helpful, but it alone may not be enough, because those people who mostly work with test tubes and restriction enzymes are unfamiliar even with such plain names. The message of this study should be highly intriguing to such people, though. Schematic illustrations of organisms in the bottom panel of Figure 5 are very intuitive for such people. It would be very helpful, if the authors could also provide such drawings for other taxonomical terms that appear in the text (such as those in the top-right panel of Figure 5).

We have now also provided the “common” names for species, and include additional drawings of species against Figure 5, as requested. The revised version provides a comprehensive description that can also be appreciated by those having limited knowledge of species diversity.

2) One of the characteristic features of the insect mushroom bodies with calyces is microglomeruli. There, dendrites of a small number of intrinsic neurons (globuli cells) contact with a presynaptic bouton of an olfactory projection neuron. Sparse coding is achieved because a globuli cell sends dendrites only to a limited number of microglomeruli. Could this be another criterion to characterize the mushroom body? Or, is it too specific to be true in only certain species?

Figure 3 shows added data on microglomeruli, employing antibodies against actin and synapsin showing layers in the stomatopod calyx packed with microglomerulus-like configurations, corresponding with those of the insect calyx.

3) Because the Stomatopoda mushroom body is situated so close to the optic lobe, and because hemiellipsoid bodies are known to integrate olfactory and visual signals, does the Stomatopoda mushroom body receive input also from the visual system? If so, does it supply the calyces, or lobes? Would it be worthwhile to compare also with the Hymenopteran brain with massive visual inputs?

The mushroom bodies in Stomatopoda are actually somewhat distant from the optic lobes, not immediately adjacent to them. There must be visual input if the MB is a sensory integrator and vision is so important in stomatopods, but this aspect of our investigation is still in progress. Intervening neuropils occur between the mushroom body columns – which flank the inner margin of the lateral protocerebrum – and the innermost optic neuropil, which is the lobula. However, the reviewer’s suggestion is a good one. We have Golgi evidence that lobula outputs extend to the reniform body, and that the reniform body then sends further connections to the mushroom body columns. At present we hesitate to be more assertive about this, as we still need some months of dye injection experiments to work out the extent of these connections.

4) It is not clear why only Hexapoda and Stomatopoda retained the ancestral ground plan while partial or massive structural loss and alteration occurred in other branches. It would be difficult to provide conclusive evidences at this stage of the study, but it would be helpful, if the authors could suggest possible mechanisms or principles. Is there any systematic correlation between the degree/tendency of structural alteration and the behavioral/functional complexity of the organisms in the respective branches?

The reviewer raises the fundamental question: if mushroom bodies are ancestral, then why have they been lost in most crustaceans? One possible explanation is that stomatopods behave like winged insects in that they are graceful and acrobatic swimmers “flying through the water” during which they use a multitude of sensory cues, including visual flow, to compute fast decisions for navigation and the pursuit of prey or conspecifics. Stomatopods establish working knowledge of their terrain; they are able to recall places (an essential property of line-trap forging, where the predatory stomatopod will visit numerous familiar hunting sites before returning to its burrow). Allocentric memory (“the organism knowing where it is in a complex sensory world”) is a property shared by many, possibly all insects that rely on visual information for visual/olfactory recognition of foraging sites, roosts, and hives. Decapods such as crabs, shrimps, and lobsters also have defined territories, but they do not engage in fast marine “flights” over substantial distances that require detailed memory of the aquatic terrain. It is also significant that species that use place memory of specific locations to accomplish certain functions, such as cleaner shrimps tending fish, or social pistol shrimps using sponges as hives, possess mushroom body-like features in their lateral protocerebra.

Very many lineages of crustaceans are wholly adapted to life in the water column, including copepods, which have the highest crustacean biomass, and on which the food pyramid rests. If mushroom bodies are pivotal for recollecting places and events associated with them, then evolving in a visually impoverished ecology speaks against retention of a metabolically costly neuropil such as a mushroom body.

Some interesting studies of stomatopod behaviors, in addition to ones cited:

· Extreme aggressiveness and territoriality: Dingle H, Caldwell RL. 1969. The aggressive and territorial behaviour of the mantis shrimp Gonodactylus bredini manning (crustacea: stomatopoda). Behaviour 33(1):115-36.

· Use of complex courtship rituals using both polarized signals and meral spots: Chiou TH, Marshall NJ, Caldwell RL, Cronin TW. 2011. Changes in light-reflecting properties of signalling appendages alter mate choice behaviour in a stomatopod crustacean Haptosquilla trispinosa. Mar Freshwater Behav Physiol 44(1):1-11.

· Hazlett BA. 1979. The Meral Spot of Gonodactylus oerstedii Hansen as a Visual Stimulus (Stomatopoda, Gonodactylidae). Crustaceana 36(2):196-198.

· Use of bluffing when freshly moulted: Adams E, Caldwell R. 1990. Deceptive communication in asymmetric fights of the stomatopod crustacean Gonodactylus bredini. Anim Behav 39(4):706-716.

· Assessment strategies for when and how to fight (evaluating the cost) (including the assessment of the size of the animal inside based on chemical cues): Caldwell RL. 1987. Assessment Strategies in Stomatopods. Bulletin of Marine Science 41:135-150.

· Use of olfactory cues when entering burrows: Caldwell, Roy L., and Karen Lamp. "Chemically mediated recognition by the stomatopod Gonodactylus bredini of its competitor, the octopus Octopus joubini." Marine & Freshwater Behaviour & Phy 8.1 (1981): 35-41.

· For a summary of behaviors: Reaka ML, Manning RB. 1981. The behavior of stomatopod Crustacea, and its relationship to rates of evolution. J Crustacean Biol 1(3):309-327.

Reviewer #2:This manuscript describes novel and interesting features of stomatopod higher brain centers. Stomatopods have highly developed visual systems and visual behaviors that may be associated with the extreme complexity of higher brain centers (hemiellipsoid bodies, which I abbreviate as HEB). The neuroanatomical data and interpretations are outstanding.However, the comparative data presented in this manuscript and available in the literature do not support the authors' primary conclusion: that the similarities between the stomatopod HEBs and insect MBs reflect an ancestral feature of the pancrustacean brain. The authors' attempts to pigeonhole the data into a case for homology creates a confusing quagmire surrounding an otherwise solid account of extreme structural variation in malacostracan HEBs.Beginning in the Introduction and Table 1, the authors flip-flop repeatedly in their assessment of features of the HEB in the studied species. The difficulty in understanding how HEBs of malacostracans differ is compounded by the questionable rationale and utility of the characteristics chosen to score MB-like structures (Table 1).

We point that prior to our manuscript insect mushroom bodies were identified according to just 3 criteria defined already in the 19^th^ Century by Flögel (1878): 1) the presence of paired lobes, 2) arising from many hundreds of cell bodies, 3) situated laterally in the forebrain.

The reviewer says that our choice of characters is of questionable utility. We disagree. The character set derives from observations of the insect brain. It was not assembled with any specific crustacean in mind but is employed in this study to test whether those characters identify an equivalent center in any of the studied crustaceans. The results show that they do.

Definitions of “character” are some of the most contentious and imprecise in evolutionary biology literature. Because it allows for uncomplicated and simple coding, our preference is Mayr’s early definition of a character: ‘any feature that may be used to distinguish one taxon from another’ (Mayr et al., 1953).

Flögel, JHL. (1878). Ueber den einheitlichen Bau des Gehirns in den verschiedenen Insecten-Ordnungen. Z. wiss. Zool 30, 556–592.

Mayr E, Lindley EG, Usinger RL. (1953).Methods and principles of systematic zoology. McGraw-Hill, NY.

On the first point: this group has published several studies on the land hermit crab Coenobita clypeatus, belonging to the eumalacostracan Infraorder Anomura (Brown and Wolff, 2012; Wolff et al., 2012 and Wolff and Strausfeld 2015). Wolff et al., 2012 states that "Comparisons of the morphology, ultrastructure, and immunoreactivity of the hemiellipsoid body of C. clypeatus and the mushroom body of the cockroach P. americana reveal in both a layered motif provided by rectilinear arrangements of extrinsic and intrinsic neurons as well as a microglomerular organization" and "…the superior part (of the HEBs) approximates a mushroom body's calyx in having large numbers of microglomeruli." These papers clearly suggest that the HEBs of C. clypeatus are similar in many regards to insect MBs.However, the authors flip flop on the similarity between C. clypeatus HEBs and insect MBs multiple times in the text.Sentence from the Introduction section:.…"although crustaceans have paired higher centers in their lateral protocerebra, those centers share none of the identifying characteristic(s) of lobed mushroom bodies as described above."

We agree that “none” was too emphatic and have clarified the revised text accordingly.

C. clypeatus HEBs have many features in common with insect MB, including orthogonal arrangements of inputs and outputs, but then later in the Introduction the authors state that C. clypeatus HEBs lack orthogonal circuitry.In Table 1, Anomura (which includes C. clypeatus) is scored as lacking a "domed center or calyces" "lobes intersected by orthogonal inputs," and "lobes partitioned as sequential domains." Table 1 thus directly contradicts the findings of Wolff et al., 2012.

The point of confusion pivots around our description of network topology. In *Coenobita*, orthogonal networks are constrained to 2-D planes or strata that occur at various depths within the domed hemiellipsoid body. This arrangement contrasts with that of a mushroom body, in which orthogonal networks are 3-D assemblies within the allantoid columns. Our revised text clarifies this distinction.

It appears that the reviewer may have misread Table 1. Anomura was not scored as lacking a domed center or calyx. Table 1 scored Anomura (exemplified by *Coenobita*) as possessing (✓(D)) “Domed center (D) or calyces (C) in lateral protocerebrum,” whereas Varunidae (Brachyura) do not. The table is now simplified and in standard format: characters are scored “1”-present, “0”-absent, “– “– no data

On the second point, the data and analysis used to generate Table 1. The traits selected to determine the identity of mushroom bodies are very confusing and their selection is not justified in the text.

The character set derives from observations of the insect brain. It was not assembled with any crustacean in mind. The revised manuscript amplifies this. The character set is clearly described.

1) Some traits are redundant (what is the difference between cell bodies supplying the lobes and supplying the calyx? The same cell population supplies both.)

Yes, the reviewer is right. It does. We have corrected this ambiguity.

2) The absence of some traits exclude others (lack of a calyx necessarily means the animal lacks cell bodies supplying the lobes, clawed distal dendrites, calyx contains orthogonal networks). This leads to differences between species that appear more dramatic than they actually are.

Lacking a “calyx” does not mean an absence of columns because, as the reviewer has just remarked in comment 1, above: the same cell body population supplies both. When globuli cell neurites have distal dendrites they contribute to a calyx. These are not dramatic differences. Thus, a calyx is not a sine qua non of a mushroom body. For example, starburst beetles, which are aquatic and anosmic (they have no antennal (olfactory) lobes) have mushroom bodies that lack calyces entirely. The lobes exclusively receive the afferent supply. Nor does the absence of a calyx obviate specific types of postsynaptic specializations. Insects lacking calyces possess lobed mushroom bodies from which parallel fibers extend branches equipped with clawed or spiny specializations (Strausfeld et al., 2009).

3) Calyx/dome are grouped together in some traits, while Lobes/dome are grouped in others. Is the HEB dome considered analogous/homologous with the MB calyces or lobes?

Yes, calyces and the HEB dome are equivalent. The arrangement of microglomeruli in both is now described in the revision.

4) Some just don't make sense. I was under the impression that orthogonal networks are characteristic of MB lobes, not calyces.

Stratified arrangements of orthogonal networks in the HEB dome pertain only to *Coenobita*, This is further clarified in this revision.

5) The characters do not take into account any aspect of behavior or sensory ecology of the animal. For example, anosmic insects would score 11/15 traits. If the MBs of insects and stomatopods are presumed to be basal, are we to assume then that Drosophila is basal to cicadas and backswimmers?

We are puzzled by this comment. Cicadas and backswimmers have undergone evolved loss and are convergent with MB morphology of odonates, which are a sister group to all other Dicondylia (see: Misof et al., 2014). Nor is it clear to us why sensory ecology should impact an interpretation of Table 1. Adding traits reflecting sensory ecology neither add nor subtract from defining a mushroom body-like organization. The absence of a calyx, for example – an adaptation seen in many aquatic species – is insignificant, particularly since the insect calyx develops later than do the lobes, a feature also cited by us in the manuscript.

6) The authors do not present data to back up scoring for some traits in some species. The Verunidae are scored as lacking "lobes or dome resolved with anti-DC0." However, Figure 5—figure supplement 2 shows the verunid Hemigrapsis nudus with DC0 labeling in the HEB. The caption for Figure 5 states that "Mushroom body lobes are recognized by their DC0 immunoreactivity across pancrustaceans." Verunids are also described as lacking orthogonal networks, but I don't understand how it's possible to determine this without Golgi impregnations to resolve individual neurons.

The legend to Figure 5—figure supplement 2 should have explained that the intense DC0-positive structure is a tract distant from the hemiellipsoid body. We have now revised this.

7) There are so many "no data" entries for most of the species, primarily due to lack of literature, that it is inevitable that stomatopods and insects will appear to share the most characters.

Yes, several species were not scored for all characters due to lack of data (such as Golgi impregnation, or immunostaining with pCaMKII and Leo). However, this does not force that stomatopods and insects will share the most characters. It might have turned out that Stomatopoda shared significantly fewer characters than those defining an insect mushroom body. In those species for which data is lacking, the absence of lobe-like extensions from domed HEB-like neuropils automatically excludes characters that define lobed/columnar mushroom-bodies.

In response to comments regarding mushroom bodies that lack calyces: the table now includes odonates.

In conclusion, this reviewer feels that factors described in the literature and discussed by the authors strongly support not homology, but convergence of the stomatopod HEBs and insect MBs. HEBs are found in a number of crustacean taxa, including some outside of the Malacostraca. In the few species studied in detail such as C. clypeatus, the HEBs share many characteristics with insect MBs, providing a basis for proposing that the complex HEBs of stomatopods through modification of HEBs. That HEBs gave rise to MB-like HEBs in one taxon of Crustacea, as simpler MBs gave rise to more complex MBs in some insects as a result of similar selective pressures for visual behavior is a far more parsimonius interpretation of the data presented and the existing literature. The rationale provided by the authors for homology is inconsistent and unconvincing.

We appreciate the reviewer’s concerns about homology versus homoplasy. Whether or not one interpretation is more parsimonious that another is very much open to debate. But essentially we agree with the reviewer: it is justified to say that mushroom body convergence is no less likely than homology and we have revised the manuscript Discussion accordingly. Criteria used for proposing phenotypic homology, such as those by Patterson, 1988 are fulfilled here. However, the reviewer is certainly not alone in advocating that stomatopod and insect mushroom bodies may be convergent and we now discuss this at length in the revised manuscript. Proof of homology between any phenotypic arrangements must eventually come from studies of their genetic organization and a recent transcriptomic study has demonstrated that detailed morphological correspondence is not necessarily indicative of genomic homology (Pankey et al., 2016).

The reviewer poses a central question: why might not the stomatopod and hexapod mushroom bodies have independently evolved from a simpler hemiellipsoid body-like ancestral state? And, might these independent evolutionary events have been driven by requirements related to superior vision, considering that many insect species and all stomatopods possess superior vision in terms of their optics, underlying optic lobe circuitry, and visual performance. We address this in the revised Discussion suggesting those aspects alone are unlikely to be coupled with the evolution of i mushroom bodies.

Reviewer #3:In their manuscript "An insect mushroom body in a crustacean brain" Gabriella H. Wolff and co-authors give an in-depth neuroanatomical and immunocytochemical analysis of a prominent neuropil in the lateral protocerebrum of mantis shrimps (Stomatopoda), that serves as second stage of the central olfactory pathway and traditionally has been known as hemiellipsoid body (HB). With this analysis the authors provide substantial evidence supporting the notion that this neuropil is in fact homologous to the well-known mushroom body (MB) of the insect brain. By additionally analyzing the expression of a protein required for learning and memory in Drosophila (DC0) in the HB in some other crustacean taxa, the authors arrive at a detailed reconstruction of the phylogenetic development of the HB/MB within Pancrustacea, the monophyletic taxon comprised of Crustacea and Hexapoda. Furthermore, the authors provide a less-detailed neuroanatomical description of a hitherto unknown neuropil in the lateral protocerebrum of crustaceans, the 'reniform body' – which in some neuroanatomical aspects is similar to the HB/MB.Overall, the study is very well executed and the main results are presented in detailed and very informative reconstruction drawings as well as in numerous images of original neuroanatomical and immunocytochemical data.There is, however, one main conceptual issue that the authors should address as well as numerous minor suggestions and comments. The main conceptual issue on which the paper hinges, is what constitutes a MB and distinguishes it from a HB – and the authors should improve their definition(s) substantially. First, there is some contradiction within the paper itself that needs to be reconciled: In Results paragraph two the authors state that a MB is identified by 12 of the 15 criteria provided in Table 1 – but they fail to identify this set of 12 criteria (the 'HBs' of stomatopods correspond to the MBs of insects in a 15 criteria according to Table 1).

This has now been clarified by providing a supplement 1 linked to Figure 1, which has now been coded appropriately. We also emphasize that the characters have been chosen from observations of insect mushroom bodies. Throughout the revision characters relate to specific illustrated examples in the main manuscript and to citing published descriptions of specific features, such as mushroom body feedback neurons (Leitch and Laurant, 1996).

While we propose phenotypic correspondence we do also draw attention to distinctions, such as the layered organization of the stomatopod calyx shown in the revised manuscript Figure 3.

The reviewer is perfectly correct in that the observed similarities beg the question whether stomatopod and insect mushroom bodies are homologous or whether they have evolved independently and are thus convergent. This is now discussed at length in the revised Discussion, emphasizing on the need for caution in interpreting phenotypic correspondence, and providing reasons for the possibility of convergent evolution.

Furthermore, later in the Results (subsection “Divergent evolution of the mushroom bodies”) the authors talk about 'mushroom body lobes' within HBs (which is a truly confusing description and concept) that apparently are identified solely by the expression of DC0.

Yes, it was confusing and we have changed this description to refer to a lobe-like extension arising from an HB-like center, specifically as the cleaner shrimp [Stenopodidae].

Even more problematic is the lack of a definition of what constitutes a 'lobed center' in the lateral protocerebrum: This is the first criterion given in Table 1 that distinguishes MBs from other neuropils in the lateral protocerebrum, which at least for Varunidae according to the authors' data include the 'reniform body' that they describe as a 'lobate center'. So why then is that not a lobed center? In fact, the term 'lobe' may be one of the least distinct anatomical terms and without a crystal-clear definition it seems impossible to exclude neuropil structures of typical HBs (or other neuropils in the lateral protocerebrum) by it. Most of the other criteria in Table 1 of the remaining 14) depend on the presence of a 'lobe' and therefore do not provide additional independent characters differentiating between HBs and MBs (it is odd that for Anomura which do not have lobed centers the density of cell bodies supplying lobes is checked). The terms 'dome' and 'calyx' also prominently appear in the Table – and they need to be rigorously defined to be useful.

We have followed this reviewer’s advice, almost to the letter. It is regrettable that during the writing of the insect brain terminology paper (Ito et al., 2014) we didn’t jettison the term mushroom body “lobe.” Mushroom bodies actually have column-like extensions from the calyces. “Lobe” is an anatomical term to describe a rounded outgrowth from a body or organ (justified for naming the optic lobe, or antennal lobe, or ear lobe!). But the term “lobe” has become engrained in the mushroom body literature, so we can’t change that now. Nevertheless, in this revision we explain that the stomatopod mushroom body’s “lobes” are best described as columns and the use of that term is now explained for describing the stomatopod mushroom body. We have also thrown out the term ‘lobe’ when discussing the reniform body and instead refer to its bundled axons as a “pedestal” that supplies discrete volumes of neuropils, referred to as named zones. The neuropil that surmounts the stomatopod’s mushroom body column is referred to as its “calyx,” the term used to denote for the neuropil surmounting the insect’s mushroom body lobes. A calyx can be cup- or dome-shaped.

We appreciate concerns regarding character linkage, a perennial issue in cladistics. Even though the designation of a character can be used for any correlated set of features (Fristrup, 1994), we have now disambiguated the characters for Table 1 (see also Figure 1—figure supplement 1, which pertains to Table 1).

Fristrup K. (1994). Character: Current Usage. In: Keywords in Evolutionary Biology (Keller EF, Lloyd EA. Eds). Harvard University Press. Cambridge, Mass.